# Systematically tracking the hourly progression of large wildfires using GOES satellite observations

Tianjia Liu[1], James T. Randerson[1], Yang Chen[1], Douglas C. Morton[2], Elizabeth B. Wiggins[3], Padhraic Smyth[4], Efi Foufoula-Georgiou[1], Roy Nadler[5], and Omer Nevo[5]

[1]Department of Earth System Science, University of California, Irvine, Irvine, CA, USA
[2]Biospheric Sciences Laboratory, NASA Goddard Space Flight Center, Greenbelt, MD, USA
[3]Science Directorate, NASA Langley Research Center, Hampton, VA, USA
[4]Department of Computer Science, University of California, Irvine, Irvine, CA, USA
[5]Google, Mountain View, CA, USA

**Correspondence:** Tianjia Liu (tianjia.liu@uci.edu)

**Abstract.** CE1 In the western United States, prolonged drought, a warming climate, and historical fuel buildup have contributed to larger and more intense wildfires as well as to longer fire seasons. As these costly wildfires become more common, new tools and methods are essential for improving our understanding of the evolution of fires and how extreme weather conditions, including heat waves, windstorms, droughts, and varying levels of active-fire suppression, influence fire spread. Here, we develop the Geostationary Operational Environmental Satellites (GOES)-Observed Fire Event Representation (GOFER) algorithm to derive the hourly fire progression of large wildfires and create a product of hourly fire perimeters, active-fire lines, and fire spread rates. Using GOES-East and GOES-West geostationary satellite detections of active fires, we test the GOFER algorithm on 28 large wildfires in California from 2019 to 2021. The GOFER algorithm includes parameter optimizations for defining the burned-to-unburned boundary and correcting for the parallax effect from elevated terrain. We evaluate GOFER perimeters using 12 h data from the Visible Infrared Imaging Radiometer Suite (VIIRS)-derived Fire Event Data Suite (FEDS) and final fire perimeters from the California's Fire and Resource Assessment Program (FRAP). Although the GOES imagery used to derive GOFER has a coarser resolution (2 km at the Equator), the final fire perimeters from GOFER correspond reasonably well to those obtained from FRAP, with a mean Intersection-over-Union (IoU) of 0.77, in comparison to 0.83 between FEDS and FRAP; the IoU indicates the area of overlap over the area of the union relative to the reference perimeters, in which 0 is no agreement and 1 is perfect agreement. GOFER fills a key temporal gap present in other fire tracking products that rely on low-Earth-orbit imagery, where perimeters are available at intervals of 12 h or longer or at ad hoc intervals from aircraft overflights. This is particularly relevant when a fire spreads rapidly, such as at maximum hourly spread rates of over $5 \, \mathrm{km \, h^{-1}}$. Our GOFER algorithm for deriving the hourly fire progression using GOES can be applied to large wildfires across North and South America and reveals considerable variability in the rates of fire spread on diurnal timescales. The resulting GOFER product has a broad set of potential applications, including the development of predictive models for fire spread and the improvement of atmospheric transport models for surface smoke estimates. The resulting GOFER product has a broad set of potential applications, including the development of predictive models for fire spread and the improvement of atmospheric transport models for surface smoke estimates (https://doi.org/10.5281/zenodo.8327264, Liu et al., 2023).

## 1 Introduction

Severe wildfire seasons in the western United States, such as in 2018, 2020, and 2021, generate large negative economic and public health impacts, displacing communities at the wildland–urban interface and inducing hazardous smoke pollution (Burke et al., 2021; Zhou et al., 2021). Following the legacy of total forest fire suppression in the 20th century, the enhanced drying of fuels from anthropogenic climate warming and a lack of prescribed burns for fuel reduction have increased the likelihood of destructive, fast-spreading megafires, such as the Creek Fire in 2020 (1537 km$^2$) and Dixie Fire in 2021 (3898 km$^2$) (Juang et al., 2022; Williams et al., 2019; Kolden, 2019; Brown et al., 2023). However, these extreme fire events, which are infrequent and outliers in terms of fire size, are often poorly characterized in statistical models of burned area or fire intensity (Wang et al., 2021; Joseph et al., 2019). As a consequence, it is important that we first understand how large fires evolve through both time and space to sufficiently model how meteorology, suppression, and fuels modulate fire spread and emissions.

Recent efforts to map the progression of fire perimeters include the Global Fire Atlas (Andela et al., 2019), GlobFire (Artés et al., 2019), Fire Events Delineation (FIRED) (Balch et al., 2020), and the Fire Event Data Suite (FEDS) (Chen et al., 2022). These products use satellite observations of fires from the Moderate Resolution Imaging Spectroradiometer (MODIS) or the Visible Infrared Imaging Radiometer Suite (VIIRS), and they cluster burned pixels or active-fire detections into individual fire events. The Global Fire Atlas, GlobFire, and FIRED use the 500 m MODIS burned-area product to map daily fire progression, while FEDS uses the 375 m VIIRS active-fire product to map 12 h fire progression. The Global Fire Atlas and GlobFire operate on a global scale, whereas FIRED and FEDS are restricted to a regional level – the contiguous United States for FIRED and California for FEDS.

Here, we improve the temporal scale of existing mapping methods for fire perimeters to hourly intervals by leveraging geostationary satellite observations from the GOES-East and GOES-West satellites. Our baseline algorithm is based on Google's initial method used to produce the wildfire layer in Google Maps (Restif and Hoffman, 2020). The wildfire layer, which updates within 30 min of GOES retrievals, displays the current perimeter of large fires based on GOES active-fire observations and aims to provide stakeholders with up-to-date information on how current fires may endanger nearby structures and lead to evacuations. To create the wildfire layer, Google Maps leverages the Google Earth Engine (GEE) cloud-based geospatial computing platform (Gorelick et al., 2017; Restif and Hoffman, 2020). GEE's petabyte-scale public data catalog maintains the GOES datasets and automatically adds and preprocesses new images as soon as they are available. GEE empowers rapid processing of a large number of data and enables the tracking of fire progression at a high temporal resolution.

In this study, we develop the GOES-Observed Fire Event Representation (GOFER) algorithm to derive the hourly fire progression of large wildfires. Our algorithm includes an optimized threshold for delineating the fire perimeter from unburned areas, parallax terrain correction for GOES images, a dynamic smoothing kernel, and scaling adjustment for early perimeters. As a test case of the GOFER algorithm, we create a product that includes hourly fire perimeters, active-fire lines, and fire spread rates for large fires that burned over 50 000 acres (202 km$^2$) in California from 2019 to 2021. A set of 28 fires met this criterion, including some of the largest (August Complex and Dixie) and most destructive (North Complex and Glass) fires in California's history. Over this 3-year span, these fires approximately accounted for 85 % of the total burned area and 77 % of all of the structures destroyed. We evaluate GOFER perimeters and active-fire lines using FEDS at 12 h intervals and validate the spatial accuracy of the final perimeter with FRAP, a fine-resolution dataset of fire perimeters derived from incident reports, remote sensing, and ground surveys. Finally, we discuss the limitations, future development, and applications of the GOFER algorithm and product.

## 2 Data and methods

### 2.1 Study region

We map the hourly progression of 28 large wildfires in California (CA) from 2019 to 2021 (Tables 1 and A1; Figs. 1 and A1). Here, we define a large wildfire as a fire that burned over 50 000 acres (202 km$^2$). The 28 wildfires include three "cross-border" fires (Slater and Devil, W-5 Cold Springs, and Tamarack) that burned across the California border into neighboring states.

### 2.2 Datasets and products

We use active-fire detections from the Advanced Baseline Imager (ABI) aboard NOAA's Geostationary Operational Environmental Satellites (GOES)-16/East and 17/West, which observe North and South America with a spatial resolution of 2 km at the Equator and a temporal resolution of 10–15 min for its full-disk view (Schmit et al., 2017; Schmidt et al., 2020). The nominal product mapping accuracy for the GOES-R Series Fire/Hot Spot Characterization product is 1 km (https://www.goes-r.gov/syseng/docs/MRD.pdf, last access: 4 January 2024). The different longitudinal positions of GOES-East (75° W) and GOES-West (137° W) yield views of the same fire from two different perspectives, generating images with two different spatial footprints for a given location. The Level-2 GOES Fire/Hot Spot Characterization product includes information on the data quality of the active-fire retrieval ("fire mask categories"), fire temperature,

**Table 1.** Metadata and GOFER-Combined summary statistics for the 28 large wildfires in California from 2019 to 2021 that burned over 50 000 acres (202 km$^2$). The area (km$^2$) refers to that of the final perimeter. Also shown are the maximum hourly concurrent (fline$_{c=0.05}$) and retrospective (fline$_r$) active-fire-line lengths, in kilometers, and the fire spread rates, in kilometers per hour, calculated from the maximum axis of expansion (fspread$_{MAE}$) and area-weighted expansion methods (fspread$_{AWE}$).

| No. | Fire name | Year | Area (km$^2$) | fline$_{c=0.05}$ (km) | fline$_r$ (km) | fspread$_{MAE}$ (km h$^{-1}$) | fspread$_{AWE}$ (km h$^{-1}$) |
|---|---|---|---|---|---|---|---|
| 1 | Kincade | 2019 | 347 | 45.7 | 25.3 | 4.9 | 2.7 |
| 2 | Walker | | 268 | 62.4 | 23 | 5.6 | 3.2 |
| 3 | August Complex* | 2020 | 4343 | 210.5 | 92.2 | 5.1 | 1.7 |
| 4 | Bobcat* | | 584 | 62.2 | 34 | 4.3 | 1.9 |
| 5 | Creek* | | 1615 | 121.7 | 52.1 | 11.3 | 4.2 |
| 6 | CZU Lightning Complex* | | 283 | 61.5 | 37.7 | 5 | 1.6 |
| 7 | Dolan | | 501 | 63.1 | 27.9 | 3.6 | 1.9 |
| 8 | Glass | | 353 | 65.5 | 29.4 | 7.6 | 3.5 |
| 9 | July Complex | | 174 | 40.1 | 27.6 | 4.6 | 1.4 |
| 10 | LNU Lightning Complex* | | 1539 | 264 | 114 | 10.5 | 1.8 |
| 11 | North Complex* | | 1344 | 126.2 | 65.6 | 9.9 | 3.1 |
| 12 | Red Salmon Complex* | | 575 | 59.7 | 26.4 | 3.1 | 1.3 |
| 13 | SCU Lightning Complex* | | 1526 | 134.7 | 62.9 | 4.8 | 1.5 |
| 14 | Slater and Devil* | | 697 | 113.9 | 39.1 | 6 | 2.9 |
| 15 | SQF Complex* | | 786 | 71.4 | 22.1 | 5.2 | 2.8 |
| 16 | W-5 Cold Springs | | 364 | 57.7 | 22.8 | 3.5 | 1.2 |
| 17 | Zogg | | 223 | 49.1 | 19.1 | 6.2 | 10.8 |
| 18 | Antelope | 2021 | 599 | 52.9 | 32.7 | 4.8 | 2.3 |
| 19 | Beckwourth Complex | | 558 | 76.3 | 31.1 | 4.7 | 1.7 |
| 20 | Caldor | | 994 | 88.7 | 42.6 | 4.3 | 2.3 |
| 21 | Dixie | | 4389 | 187.1 | 67.2 | 10.2 | 2.9 |
| 22 | KNP Complex | | 389 | 64.2 | 22.6 | 2.6 | 1.6 |
| 23 | McCash | | 406 | 67.7 | 28.6 | 2.7 | 1.5 |
| 24 | McFarland | | 567 | 62.7 | 30.4 | 4.1 | 1.4 |
| 25 | Monument | | 925 | 83.6 | 41 | 3.8 | 1.5 |
| 26 | River Complex | | 931 | 124.2 | 39.8 | 5.2 | 1.5 |
| 27 | Tamarack | | 375 | 64.2 | 23.7 | 3.7 | 3.9 |
| 28 | Windy | | 427 | 64.2 | 30 | 2.2 | 0.9 |

* Fires used in parameter optimization.

fire area, and fire radiative power (FRP), which is a proxy for fire intensity (Hall et al., 2019; Xu et al., 2010; Schroeder et al., 2010). To correct the terrain-induced parallax displacement in GOES images, we use the United States Geological Survey (USGS) 3D elevation program (3DEP) digital elevation model (DEM) at a 10 m (1/3 arcsec) spatial resolution (Archuleta et al., 2017).

We retrieve the ignition time and location of each fire from the California Department of Forestry and Fire Protection (CAL FIRE; https://www.fire.ca.gov/, last access: 4 January 2024) and InciWeb, the US interagency all-risk incident information system (https://inciweb.nwcg.gov/, last access: 4 January 2024). When CAL FIRE does not report detailed information on fires outside of its jurisdiction (i.e., on federal lands), we rely on InciWeb to fill the gap. These metadata are used to check the fire ignition time against the GOES active-fire time series and to limit the amount of GOES data spatially and temporally to process and avoid GEE computational limits.

For optimization, validation, and evaluation of GOFER, we use several datasets and products derived from higher-spatial-resolution observations: FEDS, Monitoring Trends in Burn Severity (MTBS), Fire and Resource Assessment Program (FRAP), National Interagency Fire Center (NIFC), and the CAL FIRE Damage Inspection (DINS) program. MTBS uses Landsat (30 m) imagery to map the final fire perimeter and burn severity from 1984 to the present and is available with about a 2-year lag time; MTBS maps all fires over 1000 acres (4 km$^2$) in the western US (Picotte et al., 2020). FEDS uses object-based tracking of VIIRS active fires (375 m) to map the progression of fires in California at 12 h time steps from 2012 to 2021 (Chen et al., 2022). The historical fire perimeters dataset from CAL FIRE's FRAP is the most detailed and complete dataset for California wild-

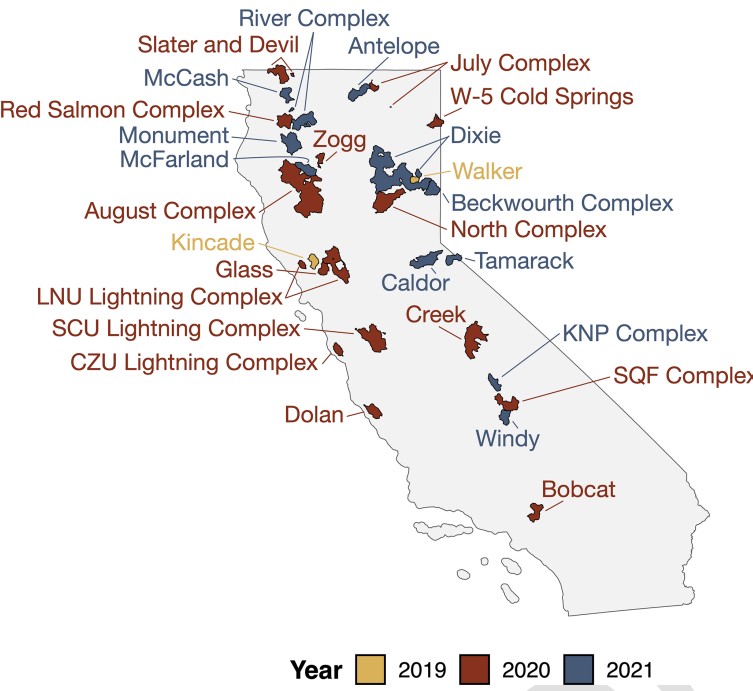

**Figure 1.** Map of the final perimeters for 28 large fires in California in the GOFER product. In total, GOFER contains 2 fires in 2019, 15 fires in 2020, and 11 fires in 2021; all if the fires mapped burned over 50 000 acres (202 km²). The footprints of the fires shown are from GOFER-Combined.

fires, which are mapped by GPS, aerial infrared observations, and other imagery (https://frap.fire.ca.gov/, last access: 4 January 2024). FRAP standardizes and combines perimeters from federal agencies (US Forest Service, Bureau of Land Management, National Park Service, and Fish and Wildlife Service). NIFC provides high-resolution intermediate perimeters derived from airborne infrared (IR) imagery by trained analysts (https://data-nifc.opendata.arcgis.com/, last access: 4 January 2024). The availability of these perimeters is sparse, varying from fire to fire, and affected by cloud cover, thick smoke, and availability of flights and coverage area. For example, almost all flights are during nighttime, and some sections of the fire may not be mapped during a particular flight. We use the IR perimeters from the US Forest Service National Infrared Operations (NIROPS) unit. After filtering the NIROPS perimeters for data quality (e.g., missing metadata and small flight coverage) and matching with GOFER perimeters by the nearest hour, our reference dataset comprises over 650 snapshots across the 28 fires. For select fires (20 of 28 fires), the CAL FIRE Damage Inspection (DINS) program database also provides the location of permanent structures inside or within 100 m of the perimeter and the level of damage sustained by each structure (accessed from the CAL FIRE Public Records Center at the GovQA portal). These data are used to calculate the number of affected and destroyed structures contained by our derived fire perimeters.

## 2.3  Using GOES active-fire detections to derive hourly perimeters

### 2.3.1  Overview of the GOFER algorithm

Restif and Hoffman (2020) show a step-by-step example of a GOES-based image-to-vector method to map fire perimeters in GEE for the 2019 Kincade Fire in California. After filtering GOES-East and GOES-West observations over a 2-week period and over an area of interest (AOI) defined as a 40 km buffer of the point location of the Kincade Fire, the GOES "fire mask codes" provided by the Fire Detection and Characterization (FDC) algorithm are remapped to fire detection "confidence" values (Table B1). This remapping arbitrarily weights the fire pixels and non-fire pixels on a continuous, interpretable scale that ranges from 0 to 1. Based on threshold tests, the GOES FDC algorithm categorizes the quality of the fire pixels as "processed," "saturated," "cloud contaminated," "high probability," "medium probability," or "low probability" (Schmidt et al., 2020). Processed and saturated codes refer to the highest-quality fire pixels, while cloud contaminated, high-probability, medium-probability, and low-probability codes refer to lower-quality fire pixels that may be false alarms. For each satellite, the maximum fire detection confidence is calculated from GOES images retrieved within the input temporal limits, and the GOES-East and GOES-West maximum fire detection values are combined by taking the minimum. Next, the combined GOES fire detection confidence map is smoothed using a

2 km square kernel. A confidence threshold of 0.6 is applied to mask low-confidence areas, and the image is then converted into a vector at a spatial resolution of 200 m. As the resulting vector retains unnatural edges from the footprint of the image pixels, the vector is simplified within a maximum error of 500 m, thereby smoothing any jagged edges.

Here, we expand and improve the Restif and Hoffman (2020) method by adding four optimizations or adjustments in our GOFER algorithm: (1) dynamic smoothing kernel size, (2) early-perimeter adjustment, (3) parallax terrain correction, and (4) confidence threshold optimization. Specifically, we reduce the arbitrary selection of parameters by optimizing against perimeters derived from high-resolution satellite imagery, increase the geolocation accuracy of GOES fire pixels with a parallax terrain correction, and improve the mapping of early perimeters. In Fig. 2, we pictorially depict the steps to produce the final perimeter of the 2020 Creek Fire as an example.

In step 1, following Restif and Hoffman (2020), we assign GOES-East and GOES-West fire mask codes as fire detection confidence values (Table B1) and (for each satellite) calculate the maximum fire detection confidence over the temporal stack of images from ignition to the end hour. For the Creek Fire, the average spatial resolution is about 3.2 km for GOES-East and 2.5 km for GOES-West, calculated from the GOES pixel area within a bounding box covering the fire's extent. Due to the different pixel orientations and resolutions of the GOES-East and GOES-West grids, we overlay them to create a combined grid at a downscaled spatial resolution. The combined grid is heterogenous in pixel size with an area-weighted spatial resolution of 1.7 km. The spatial resolution of the combined grid is later used in step 3 to determine the kernel radius to smooth the fire detection confidence image.

In step 2, we apply scaling factors from the early-perimeter adjustment to the stack of hourly fire detection confidence images. The early-perimeter adjustment ensures that perimeters are formed at the start of a fire despite dilution from neighborhood smoothing in step 3 and despite the possible absence of high-fire-confidence pixels to overcome the confidence threshold applied in step 4. We combine the GOES-East and GOES-West maximum fire detection confidence by taking the average. We also correct the terrain-induced parallax displacement in each satellite. Due to the elevation and location of the fire relative to the satellite's viewing angle, the GOES-observed fire pixels are displaced from their actual location; displacements are greater for fires at high elevations and located toward the edge of the GOES disk. The early-perimeter adjustment and parallax correction are needed steps to improve the respective temporal and spatial accuracy of the perimeter, but they are not accounted for in Restif and Hoffman (2020).

In step 3, we smooth the values using a square kernel with a radius equal to the area-weighted spatial resolution of pixels within the AOI. Restif and Hoffman (2020) set an arbitrary kernel size of 2 km, whereas our dynamic calculation

of the kernel size accounts for the heterogenous pixel size of the combined grid. Using the kernel to apply a neighborhood mean, the smoothing transforms the fire detection confidence values into a continuous gradient and removes "blockiness" at the edges.

In step 4, we apply a threshold mask of 0.95 to the smoothed confidence values. Restif and Hoffman (2020) arbitrarily set the confidence threshold to 0.6, while we optimize for the confidence threshold, as discussed in Sect. 2.3.3. In addition, Restif and Hoffman (2020) use a spatial resolution of 200 m for the intermediate image with the smoothed fire detection. We opt for a higher spatial resolution of 50 m to reduce blockiness at the edges of the polygon formed in step 5. At a coarser resolution, the edges of the polygon are more staircase-like, mirroring the pixel edges of the raster.

In step 5, the image is converted to a polygon that represents the fire perimeter. To further smooth the geometric complexity induced by the image-to-vector conversion and reduce the file size of the polygon, we simplify the polygon with a maximum error margin of 100 m, which is in a 2 : 1 ratio with the spatial resolution of the smoothed confidence image. This ratio is similar to Restif and Hoffman (2020), who set the maximum error margin to 500 m, versus 200 m, for the smoothed fire confidence image.

In addition to the combined GOES method, we also create perimeters and related fire metrics solely using GOES-East imagery or GOES-West imagery to test the efficacy of using just one satellite. We separately optimize the confidence threshold and parallax adjustment factor and calculate the smoothing kernel size and early-perimeter adjustment for each GOFER version. Hereafter, we refer to the three GOFER versions as GOFER-Combined, GOFER-East, and GOFER-West. For this study, GOFER-Combined uses GOES-16 and GOES-17 observations, GOFER-East uses only GOES-16 observations, and GOFER-West uses only GOES-17 observations. We note that GOES-17 was replaced by GOES-18 in early 2023.

### 2.3.2 Preprocessing: input metadata dictionary

In the preprocessing stage, we create a metadata dictionary of input values for each fire (Fig. 3). Here, "dictionary" refers to the data structure in code stored as "key" and "value" pairs, where the keys, or user-specified words, are used to retrieve the corresponding values. In particular, we set temporal and spatial constraints for calculating fire progression, i.e., the start and end time bounds and AOI polygon. For the start time, we use the ignition time as reported by CAL FIRE, when possible, or InciWeb and round down by hour (e.g., 06:37 to 06:00 LT). However, GOES can detect active fires prior to the ignition time for some fires – mainly lightning-caused fires; for such cases, we set the hour of the earliest GOES active-fire detection as the start time. We set the end time as the hour with the last GOES active-fire detection that occurs within a few days of previous detections, provided

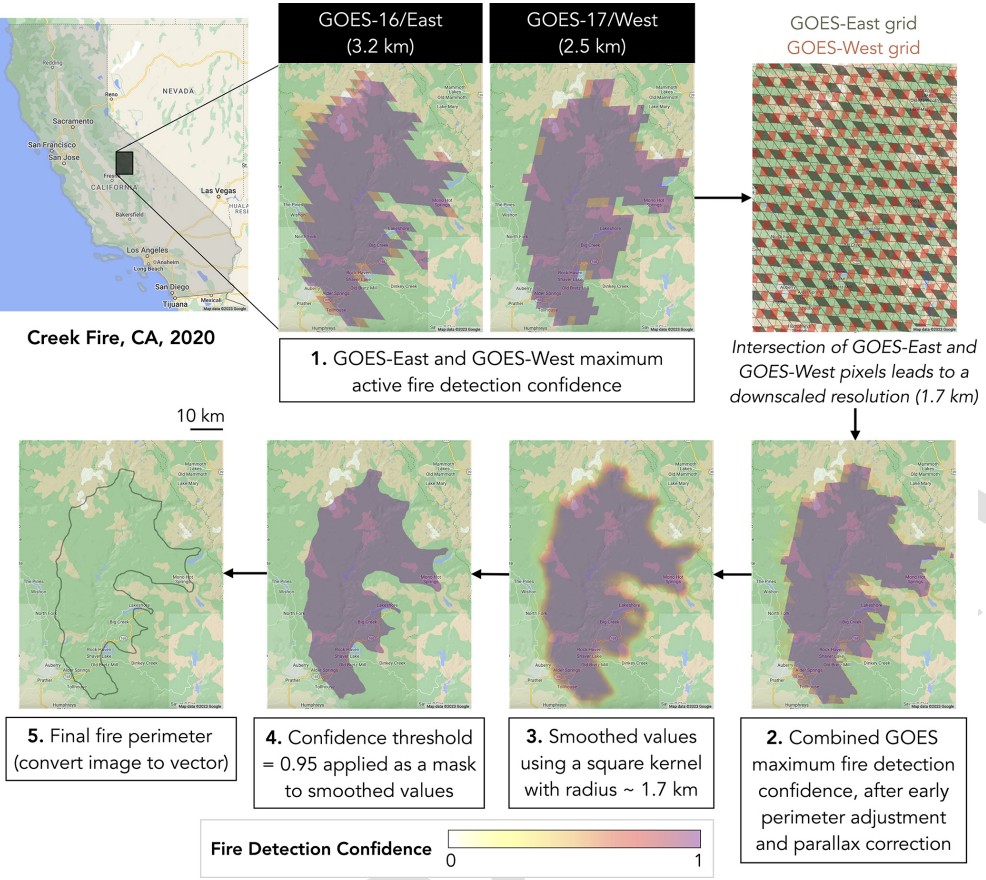

**Figure 2.** Pictorial depiction of the GOFER workflow used to produce fire perimeters from GOES active-fire detections in Google Earth Engine. The gray shaded area represents the state of California; the black box shows the location of the Creek Fire in 2020. This example shows the workflow for producing the final fire perimeter of the Creek Fire and uses all GOES images from the hour of ignition to the last fire detection. The GOES nominal spatial resolution is 2 km at the Equator but varies based on the pixel's location relative to the longitudinal position of the GOES satellite; the GOES resolutions inset are specific to the Creek Fire. The background map data are from ©Google Maps 2023, rendered on the Google Earth Engine platform.

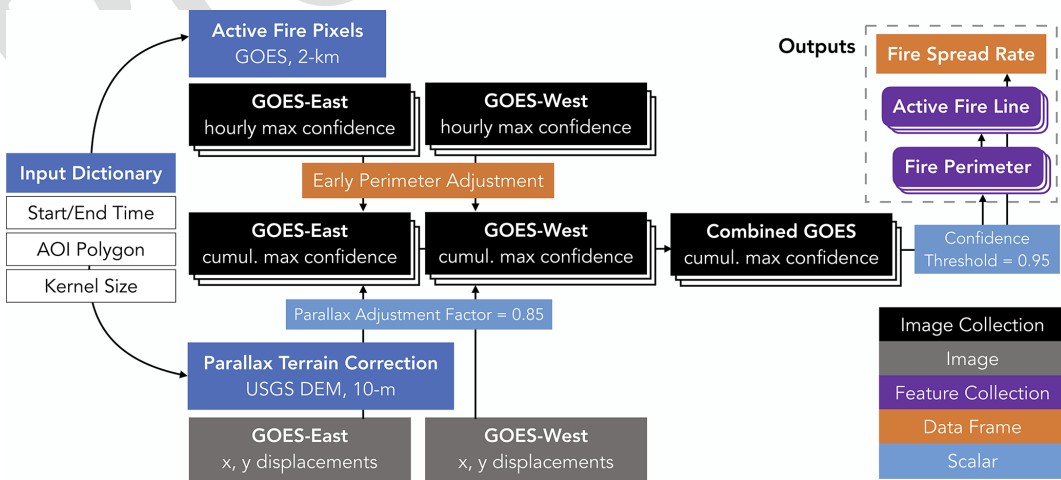

**Figure 3.** Overview of the GOFER workflow used to produce the GOES-derived fire perimeters and ancillary fire metrics (active-fire line and fire spread rate). The confidence threshold and parallax adjustment factor values are optimized using the 10 largest wildfires in California in 2020. The dark blue boxes are headings to denote the different input data.

that the fire has converged to close to its final size recorded by CAL FIRE or InciWeb. This is an approximate estimate of the end time, as a later quality control step sets the end hour as when the fire perimeter last expanded (Sect. 2.3.4). For the AOI polygon, we start with the CAL FIRE or Inci-Web ignition coordinates and expand to a simple rectangle or polygon that includes the footprint of GOES active-fire detections related only to that fire.

### 2.3.3 Processing: development, optimization, and improvements

In the processing stage, we implement the four optimizations or adjustments in the GOFER fire perimeter mapping method: (1) dynamic smoothing kernel size, (2) early-perimeter adjustment, (3) parallax terrain correction, and (4) confidence threshold optimization (Fig. 3). For GOFER-East and GOFER-West, we separately optimize the confidence threshold and parallax adjustment factor and calculate the smoothing kernel size and early-perimeter adjustment. Software details specific to GEE are provided in Appendix B1.

#### Dynamic kernel size

As described above, the radius of the square kernel used for smoothing is calculated as the spatial resolution of the combined GOES grid within the AOI polygon. Opting for a dynamic kernel size, instead of a static value of 2 km used in Restif and Hoffman (2020) for example, allows the algorithm to be applied more effectively to fires outside California. The GOES spatial resolution per pixel decreases away from the Equator and toward the edge of the disk (Fig. B1). The kernel size is calculated in the preprocessing stage and added to the input metadata dictionary.

#### Early-perimeter adjustment

Some fires smolder at low intensity, leading to low-confidence detections at the beginning of their lifetime. Consequently, the GOFER algorithm fails to output these early perimeters, as the confidence values do not meet the required threshold. We add an adjustment to "anchor" the first perimeter at or close to the first available GOES fire detection by scaling the fire detection confidence. For each hour, the scaling factor is calculated as the maximum of all values in the cumulative maximum confidence image up to that hour. The scaling factor ranges from 0 to 1, where 1 indicates no scaling; however, we set the minimum scaling factor to 0.1 to prevent overinflation of early perimeters (Fig. 4b). To perform the early-perimeter adjustment, the hourly maximum confidence is divided by the scaling factor.

#### Confidence threshold optimization and parallax correction

Next, we simultaneously optimize for the confidence threshold and parallax adjustment factor. The confidence threshold applies a mask to the smoothed fire detection confidence and removes values lower than the threshold. The parallax adjustment factor ranges from 0 to 1 and is multiplied by the parallax displacement in the $x$ and $y$ components; this range allows us to test the efficacy of the parallax correction on the spatial accuracy of the final perimeter. The parallax correction algorithm is a function of the terrain elevation, the height of the satellite, the longitudinal position of the satellite, Earth's semi-minor and semi-major axes, and Geodetic Reference System 1980 (GRS 80) eccentricity (Spestana et al., 2022). We use the USGS 10 m 3DEP DEM as input. The displacement is smoothed using the same square kernel for smoothing the GOES fire detection confidence. This prevents extreme displacements of smaller 10 m pixels within a coarse GOES pixel that may contain large variations in elevation.

For optimization, we test the confidence threshold in increments of 0.01 from 0.75 to 0.99 and the parallax adjustment factor in increments of 0.05 from 0 to 1 (Fig. 4a). For each combination of the tested confidence threshold and parallax adjustment factor, we calculate the IoU of the GOFER and MTBS final perimeters. The IoU, or Jaccard index, is a common metric for evaluating spatial accuracy against ground truth data in object detection. Here, the IoU is calculated as the area of overlap over the area of union using the fire perimeters, in which an IoU of 0 indicates no agreement and an IoU of 1 indicates perfect agreement. We take the optimal values at the maximum IoU (Table B2). As this process is computationally intensive, the parameter search uses the 10 largest fires in California in 2020, a subset of the 28 fires in this study.

### 2.3.4 Post-processing: quality control

In the post-processing stage, we undertake quality control of the hourly perimeters. For each time step, we ensure that the perimeter is spatially inclusive of previous perimeters by taking the union of that perimeter and previous perimeters. We set the last time step as when the perimeter last grew and remove extraneous perimeters.

### 2.4 Derived fire metrics

From the GOES-derived progression perimeters, we compute several key fire metrics, including the diurnal cycle of the fire growth in units of area ($km^2$), active-fire-line length (km), and fire spread rate ($km\,h^{-1}$). Figure 5 illustrates the methods for calculating the active-fire line and fire spread rate. We use simple polygons to depict hypothetical perimeters at time steps $t = 0$ to $t = 2$, or from ignition ($t = 0$) to the current hour ($t = 1$) to the next hour ($t = 2$). The ignition point is defined as the centroid of the perimeter at $t = 1$.

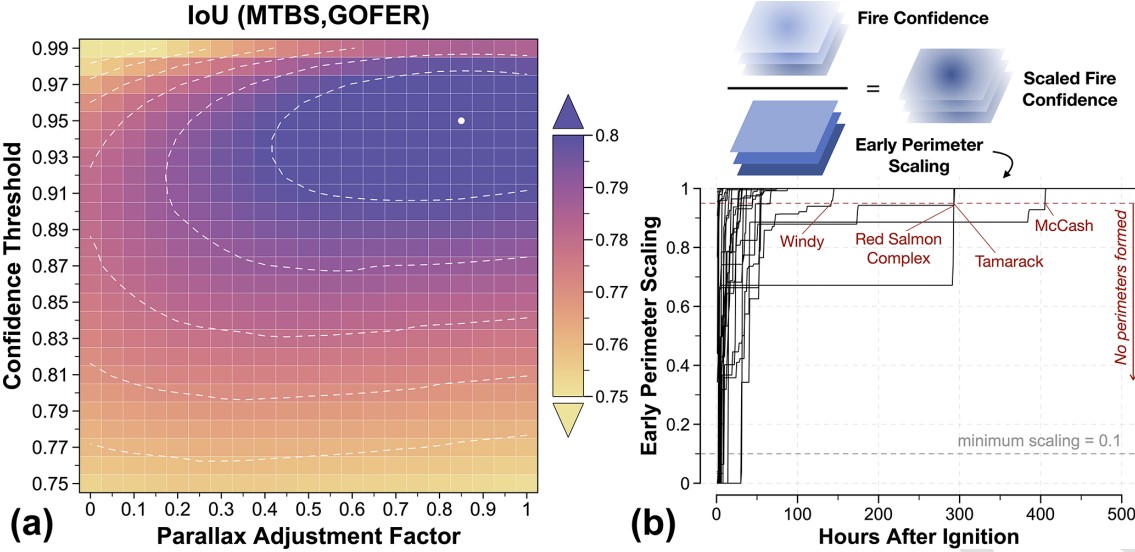

**Figure 4.** Parameter optimization and early-perimeter adjustments for deriving the GOFER-Combined fire progression perimeters. **(a)** Parameter optimization of the confidence threshold and parallax adjustment factor. The optimization is based on the Intersection-over-Union (IoU) of GOFER and MTBS perimeters at the final extent of the fire, averaged across the 10 largest CA fires in 2020. At the maximum IoU, the optimized confidence threshold is 0.95, and the parallax adjustment factor is 0.85. **(b)** Early-perimeter scaling. Adjustment for the fire confidence of early perimeters is shown as a function of hours after ignition, with individual lines depicting each of the 28 largest fires in CA from 2019 to 2021. The hourly fire confidence is divided by the early-perimeter scaling to calculate the scaled fire confidence. The minimum scaling, denoted by the dashed gray line, is set at 0.1 to prevent overly inflating early perimeters. The optimized confidence threshold of 0.95 for GOFER-Combined is denoted by the dashed red line. When the early-perimeter scaling is lower than the confidence threshold, a perimeter cannot be formed without any adjustment. The four fires depicted would have had their first perimeter formed hundreds of hours after ignition without the early-perimeter scaling.

### 2.4.1 Active-fire line

We identify the active-fire line in two ways, as either the "concurrent" or the "retrospective" active-fire line. Both active-fire-line lengths are in units of kilometers.

The concurrent active-fire line ($\text{fline}_c$) is defined as the segments of a given fire perimeter that intersect with active-fire detections of that hour above a certain threshold. For each hour, we separately output $\text{fline}_c$ at confidence thresholds $c$ of 0.05, 0.1, 0.25, 0.5, 0.75, and 0.9; this set of $\text{fline}_c$ at varying
thresholds allows us to progressively narrow down perimeter segments with the most intense burning. A lax threshold, such as $\text{fline}_{c=0.05}$, uses most of the active-fire detections during that hour, whereas a strict threshold, such as $\text{fline}_{c=0.9}$, only uses high-confidence detections to create the
hourly GOES fire perimeters. The $\text{fline}_{c=0.05}$ is most comparable to active-fire lines in other satellite-derived products such as FEDS, which uses all active-fire pixels intersecting with the perimeter. $\text{fline}_c$ with stricter thresholds correspond to areas with higher fire intensity. We convert the perimeters
from polygons to linestrings and use a buffer of 100 m around the perimeter to extract intersecting active-fire pixels with a fire detection confidence above the defined threshold.

The retrospective active-fire line ($\text{fline}_r$) is defined as the segments of a given fire perimeter that leads to growth in
the next hour's perimeter. Because of this strict definition,

the $\text{fline}_r$ is generally shorter than the $\text{fline}_c$ that is defined using low-confidence thresholds (e.g., $c = 0.05$), as the latter may include segments of the perimeter that may be actively burning but have not yet expanded during that hour.

For both the $\text{fline}_c$ and $\text{fline}_r$, we consider the perimeter as
"growing" in a given time step if the active-fire-line length is $> 0$ and "dormant" otherwise. We fill in dormant time steps with the most recent $\text{fline}_c$ prior to that time step and the most immediate $\text{fline}_r$ after that time step.

In general, the $\text{fline}_c$ can be calculated in near-real time
along with perimeters and is most useful for identifying potential areas of spread along the perimeter and testing predictive models of future fire growth. The set of $\text{fline}_c$ at different confidence thresholds can be used in tandem to identify the least to most probable segments of future perime-
ter expansion. Whereas the $\text{fline}_c$ is not necessarily associated with perimeter expansion (e.g., indicates smoldering or natural/human barriers), the $\text{fline}_r$ requires knowledge of future perimeters but offers a more precise estimate of where the perimeter expanded. The $\text{fline}_r$ is a stricter definition of
the active-fire line, more similar in length to $\text{fline}_c$ at high-confidence thresholds. The $\text{fline}_r$ can be used for retrospective analysis to assess the drivers and barriers of fire growth.

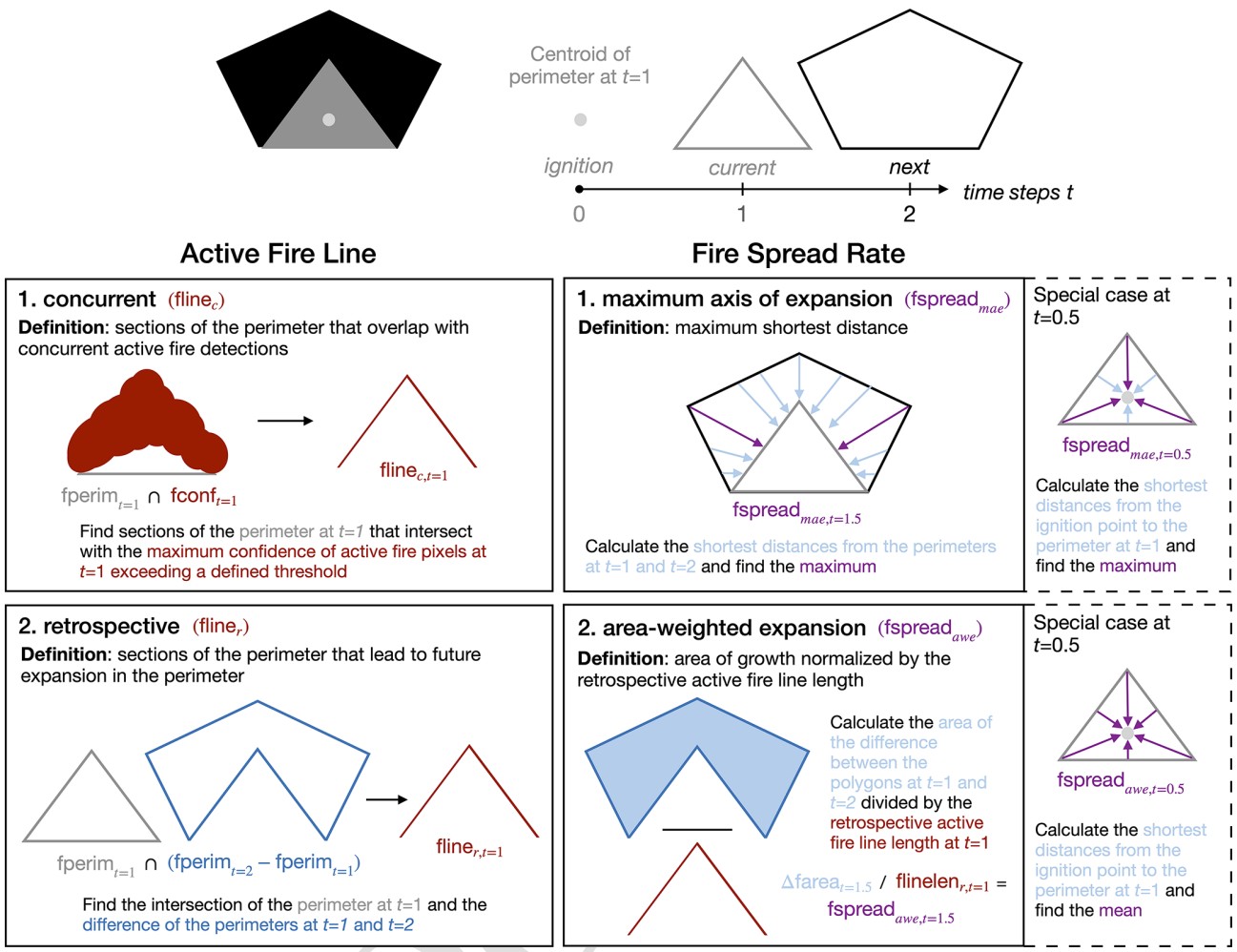

**Figure 5.** Pictorial overview of definitions for delineating active-fire lines and calculating the fire spread rates. A simplified representation of perimeters is shown with the ignition point at time step $t = 0$, the current perimeter at time step $t = 1$, and the next perimeter at time step $t = 2$. We use a (1) concurrent and (2) retrospective method for delineating active-fire lines. The concurrent method relies on the intersection between the fire perimeter and concurrent active-fire detections, whereas the retrospective method uses future perimeters to determine which portion of the current perimeter leads to a growth in area. We also define the fire spread rate from the (1) maximum axis of expansion (MAE) and (2) area-weighted expansion (AWE). The MAE fire spread rate is calculated from the maximum shortest distance between two perimeters, whereas the AWE fire spread rate is calculated as the area of growth normalized by the retrospective active-fire-line length.

### 2.4.2 Fire spread rate

To quantify the apparent horizontal expansion of the fire perimeter, we define the fire spread rate, in kilometers per hour, in two ways, as either the maximum axis of expansion (fspread$_{MAE}$) or the area-weighted expansion (fspread$_{AWE}$), between two hourly time steps. Similar to the approach in Benali et al. (2023), fspread$_{MAE}$ represents the partial fire spread along the longest axis of expansion, whereas fspread$_{AWE}$ represents the overall fire spread. While the fire perimeter and active-fire line describe the state of the fire at the end of the hour ($t = 1, 2, 3\dots$), the fire spread rate, along with the growth in the area, describes the change in state between consecutive perimeters; thus, we set these latter variables at the half hour ($t = 0.5, 1.5, 2.5\dots$). For example, the fire spread rate at $t = 1.5$ is calculated from the perimeters at $t = 1$ and $t = 2$.

The MAE fire spread rate (fspread$_{MAE}$) is calculated as the maximum shortest distance between consecutive perimeters. For the fspread$_{MAE}$ at $t = 1.5$, we calculate the shortest distance from the perimeter at $t = 1$ outward to all pixels within a search radius of 100 km. We then extract the maximum distance value within the area of growth between the perimeters at $t = 1$ and $t = 2$. In the case where there is no previous perimeter, such as the fspread$_{MAE}$ at $t = 0.5$, we set the previous perimeter, at $t = 0$, as the centroid of the perimeter at $t = 1$. In the case of fires merged from smaller fires, we disaggregate multipolygons into separate polygons and search for new ignitions or polygons that do not overlap with the

**Table 2.** Variables in the GOFER product.

| Name | Short name | Units |
|------|-----------|-------|
| **Global variables** | | |
| Fire name | fname | |
| Fire year | fyear | |
| **End-of-hour variables ($t = 1, 2, 3 \ldots$)** | | |
| Hours after ignition, end of hour | timestep | h |
| UTC time | tUTC | |
| Local time, with daylight savings | tLocal | |
| Local time, without daylight savings | tLocalGMT | |
| Area within fire perimeter | farea | km$^2$ |
| Area within fire perimeter, as a percentage of the final area | fareaPer | % |
| Active-fire-line length (concurrent) | cflinelen | km |
| Active-fire-line length (retrospective) | rflinelen | km |
| Length of the perimeter | fperim | km |
| State of the fire | fstate | 0 = dormant, |
| | | 1 = active |
| **Half-hour variables ($t = 0.5, 1.5, 2.5 \ldots$)** | | |
| Hours after ignition, half hour | timestep_hh | h |
| Growth in fire-wide area | dfarea | km$^2$ |
| Fire spread rate (MAE) | maefspread | km h$^{-1}$ |
| Fire spread rate (AWE) | awefspread | km h$^{-1}$ |

previous perimeter. If no overlap exists for a polygon, we add the centroid of that polygon to the previous perimeter.

The AWE fire spread rate (fspread$_{AWE}$) is calculated as the fire-wide growth in area divided by the retrospective active-fire-line length. The fspread$_{AWE}$ at $t = 1.5$, for example, is calculated as the change in area, in square kilometers, from the perimeter at $t = 1$ to the perimeter at $t = 2$ divided by the fline$_r$ length at $t = 1$. The calculation of fspread$_{AWE}$ for the special case of when there is no fline$_r$ at ignition, or during the time step just prior to the first formed perimeter (here depicted as $t = 0$), is similar to that for fspread$_{MAE}$, except here we take the average rather than the maximum.

### 2.4.3  GOFER product structure and variables

The GOFER product for the 28 large CA fires contains hourly fire perimeters, active-fire lines, and fire spread rates for three GOFER versions: GOFER-Combined, GOFER-West, and GOFER-East (Liu et al., 2023). Table 2 describes variables contained in the GOFER product. We provide shapefiles (.shp) of the perimeters and concurrent and retrospective active-fire lines and a summary table (.csv) of all end-of-hour and half-hour variables. End-of-hour variables record the state of the fire each hour, whereas half-hour variables record the change in the fire between 2 consecutive hours.

### 2.5  Validation and evaluation

In our framework, the spatial accuracy of the perimeters directly affects that of the active-fire lines and fire spread rates, both of which are derived from the perimeters. Due to limitations in high-resolution reference data, we focus on the validation of the perimeters with FRAP and NIROPS and evaluation of active-fire lines with comparisons to FEDS here.

To validate the spatial accuracy of the GOFER perimeters, we calculate the IoU of GOES and FRAP final perimeters. We compare this to the IoU of the FEDS and FRAP final perimeters. Further, to quantify the spatial error between the GOFER and FRAP final perimeters, we calculate "breakpoints" in the distribution of shortest distances from the GOFER perimeter to the FRAP perimeter. These breakpoints are defined by the mean and several percentiles, including the median and maximum, and the magnitude of these distances represents the deviation of the GOFER perimeter from the ground truth. This spatial error is induced by a combination of coarse spatial resolution, geolocation error, and missing fire detections in GOES. Our analysis is similar to the evaluation described in Ben-Haim and Nevo (2023) for GOES-derived fire perimeters but incorporates both false positives and false negatives in one metric. We use the fire structure status dataset from CAL FIRE as another way to validate the GOFER perimeters by calculating the number of affected and destroyed structures contained by the final perimeter. Specifically, this evaluates omission error, as damaged and de-

stroyed structures should be located within the final perimeter.

To validate the temporal progression of GOFER perimeters, we use NIROPS perimeters derived from airborne IR imagery. We track the change in fire area between snapshots and the cumulative fire area relative to the final fire size. Because NIROPS perimeters are relatively sparse and almost all during nighttime, we additionally evaluate the performance of GOFER relative to FEDS over each fire's lifetime to check when the GOFER perimeters are relatively stable with respect to spatial accuracy. This provides a partial test of GOFER's performance. To do so, we track the IoU of GOFER and FEDS perimeters, as well as the fraction of false positives and false negatives, at 12 h intervals. As a caveat, the perimeters and active-fire lines in FEDS are labeled as day or night, and the exact timing of the overpass, which can differ by more than half an hour from day to day, is not provided with the product. Based on the approximate 01:30/13:30 overpasses for VIIRS, we compare FEDS to GOFER at 02:00/14:00.

We evaluate the GOFER concurrent active-fire lines at the different confidence cutoffs (0.05, 0.1, 0.25, 0.5, 0.75, and 0.9) compared to the FEDS active-fire lines. We determine which cutoff leads to the highest agreement with the FEDS active-fire lines. However, the GOFER and FEDS algorithms still inherently differ. FEDS can take advantage of the higher spatial resolution of 375 m VIIRS detections to identify fire locations more accurately than GOES, whose raw active-fire detections can lead to large biases due to its much coarser spatial resolution. Thus, the different GOES confidence cutoffs provide a range of concurrent active-fire lengths loosely tied to varying levels fire intensity at the fire front. As another check, we calculate the aggregate 12 h retrospective active-fire-line lengths for both GOFER and FEDS perimeters.

## 3 Results and discussion

### 3.1 Evaluating the accuracy of the GOFER fire progression perimeters

Figure 6 shows the hourly GOFER-Combined perimeters for the 10 largest CA fires in 2020 that were used to optimize the confidence threshold and parallax adjustment factor (Fig. 4). The optimized confidence threshold is 0.95 for GOFER-Combined, higher than GOFER-East (0.76) and GOFER-West (0.83). The optimized parallax adjustment factor ranges from 0.8 to 1 among the GOFER versions, suggesting that the parallax correction is a needed step to improve the spatial accuracy of GOES active-fire pixels (Table B2). Specifically, for GOFER-Combined, the mean IoU for the 10 fires is 0.78 when no parallax adjustment is applied (adjustment factor $= 0$), compared to 0.81 at the optimized adjustment factor of 0.85 (Fig. 4a). The effect of the parallax correction is apparent for the Creek Fire, which was located on mountainous terrain at a mean elevation of about 1.8 km above sea level. Its uncorrected final perimeter deviates from the FRAP perimeter on the northern and eastern edges, lowering the IoU by 0.09. (Fig. B3).

We evaluate the spatial accuracy of GOFER fire perimeters at the final time step compared to FRAP, on select days compared to NIROPS, and at 12 h intervals compared to FEDS. For the 28 large fires, the mean IoU of GOFER and FRAP perimeters is 0.77 for GOFER-Combined, 0.67 for GOFER-East, and 0.75 for GOFER-West (Table C1). In general, the lower IoU for GOFER-East, due to the coarser resolution of GOES-East compared with GOES-West in California, suggests that GOES-West drives the improved spatial accuracy of the GOFER-Combined perimeters. Because of the larger smoothing kernels used in GOFER-East, the perimeters generally smooth over burned peninsula and inlet-type features where the fire conforms to the sinuous, mountainous terrain (Fig. C1, Tables B2 and C2).

The overall temporal progression of the cumulative change in fire-wide area in GOFER agrees well with NIROPS. For example, for hours that NIROPS perimeters are available, we find a strong correlation between the change in fire area ($r = 0.86$, RMSE $= 52.8$ km$^2$) between NIROPS snapshots and fractions of final fire size ($r = 0.99$, RMSE $= 0.05$) from GOFER-Combined and NIROPS (Fig. C2). The high RMSE in the change in fire area mainly stems from a few instances of high bias between some snapshots in complex fires. The median absolute bias is $6.7$ km$^2$, while the mean absolute bias is $16.7$ km$^2$. GOFER has a median positive bias of 0.02 in the fractions of final fire size, suggesting that perimeter growth accumulates slightly earlier for GOFER than for NIROPS. As a caveat, NIROPS does not fully map the fire for some snapshots, so some areas of active growth may be missing. The discrepancies may also indicate that GOFER is unable to pick up small increments of growth later in the fire's lifetime when fire front is less active.

Figure 7 compares the Creek Fire progression mapped by GOFER-Combined and FEDSv2. Although the FEDS perimeters are more detailed, GOFER fills in gaps in the fire progression when the fire spreads rapidly ($< 50$ h after ignition for the Creek Fire), thereby providing insights into the fire's behavior when it is most explosive. We also compare the IoU of GOFER to FEDS relative to FRAP for 25 large fires, which excludes the three cross-border fire that are not fully mapped in FEDSv2 (Sect. 2.1).

The IoU for FEDS is 0.83, higher than the IoU of 0.77 for GOFER-Combined, 0.68 for GOFER-East, and 0.76 for GOFER-West (Fig. 8a, Table C1). This discrepancy is reasonable considering the higher spatial resolution of the input active fires in FEDS (375 m) compared with GOFER (GOES-East: 3.1–3.6 km; GOES-West: 2.5–2.7 km; GOES-Combined: 1.6–1.7 km) (Table B2). In addition, the average IoU for the 10 megafires in 2020 that we used to optimize parameters is similar to the IoU for the 7 megafires in 2021 (e.g., the IoU for GOFER-Combined and FRAP is 0.8 for 2020 fires and 0.78 for 2021 fires). The lack of a significant

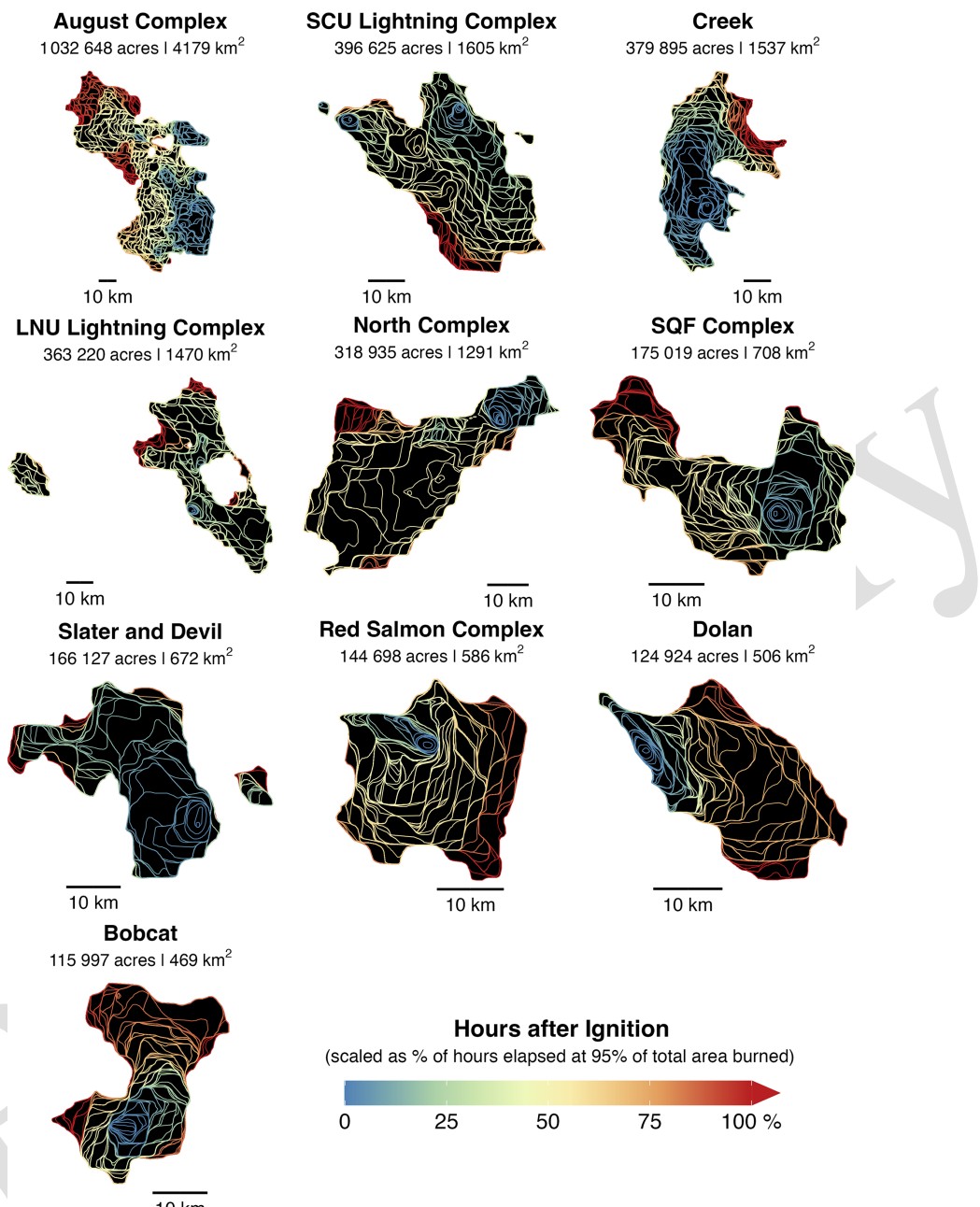

**Figure 6.** Maps of the GOFER-Combined hourly fire progression perimeters of the 10 large fires that burned over 100 000 acres in CA in 2020. For each fire, the official burned area (in acres and square kilometers) from CAL FIRE is inset. Cooler colors represent time steps early in the fire's lifetime, whereas warmer colors represent time steps later in the fire's lifetime. The time steps are normalized across fires and expressed as the percentage of hours elapsed relative to the time step at 95 % of total area burned.

drop in the IoU suggests that our parameters are not over-tuned to those 10 fires in 2020.

Using FEDS perimeters as 12 h references, we find that the IoU of GOFER and FEDS begins to stabilize around 100 h after ignition (IoU > 0.6) (Fig. 8b). We find a similar pattern using NIROPS perimeters. At < 100 h, the fires are small and, therefore, harder to map accurately at GOES resolution, as any small shift in the perimeter can lead to

a sizable decrease in the IoU. Another reason for the low IoU < 100 h after ignition is that some fires required extensive early-perimeter adjustment to scale the fire detection confidence and output a rough estimate of these early perimeters. In particular, the fraction of false positives is higher than that of false negatives close to ignition due to overinflation in GOFER early perimeters. At the cost of spatial accuracy, we anchor the first perimeter close or at the time step with the

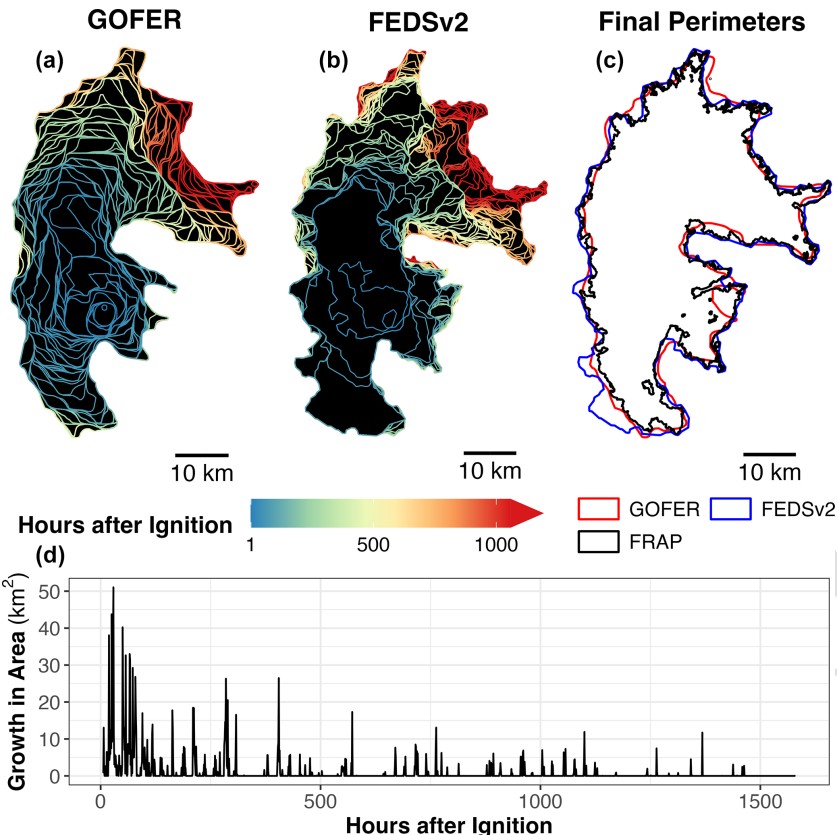

**Figure 7.** Spatiotemporal progression and comparison of the 2020 Creek Fire. Maps of the hourly GOES-derived GOFER-Combined progression **(a)**; 12 h VIIRS-derived FEDSv2 progression **(b)**; and comparison of the GOFER-Combined, FEDSv2, and FRAP final perimeters of the Creek Fire **(c)**. **(d)** Time series of the hourly growth in area for the Creek Fire from GOFER-Combined.

first GOES active-fire detection. If the scaling factor from the early-perimeter adjustment is lower than the confidence threshold, this indicates that a perimeter could not be formed. In extreme cases, such as the Windy, Tamarack, Red Salmon Complex, and McCash fires for GOFER-Combined, we see this inability to form an initial perimeter hundreds of hours after ignition (Fig. 4b).

Certain conditions or features lower the spatial accuracy or IoU – namely, the obscuration of the satellite view due to clouds and heavy smoke, location of a fire along a coastal boundary or waterbody, and presence of unburned islands and narrow burn scar features. Of the 28 fires, the main outlier is the July Complex Fire, which has a low IoU of 0.44 for GOFER-Combined and 0.48 for FEDS (Table C1). Because active-fire detection relies on discovering instantaneous thermal anomalies, clouds or thick smoke could prevent both satellite sensors from detecting active fires. On the other hand, burned-area mapping, such as in MTBS or FRAP, incorporates a time series of pre-fire to post-fire land cover changes, so is possible to infer burned area during very cloudy or smoky periods from later observations. In addition, GOFER tends to underestimate the perimeter extent for fires that hug the coast (e.g., Dolan) or have narrow burn

scar features (e.g., LNU Lightning Complex) (Figs. 1 and 6). The neighborhood smoothing in GOFER yields low-fire-detection-confidence values along the edge of the coast and around narrow burn scars, which shrinks the perimeter and can even lead to fragmentation (e.g., SCU Lightning Complex). This issue is more acute in GOFER-Combined, which uses a higher, and therefore stricter, confidence threshold than GOFER-East and GOFER-West. As such, we observe a lower percentage of damaged and destroyed structures within GOFER-Combined (92 %) and GOFER-East (93 %) perimeters compared with GOFER-West (99 %), signifying that the higher omission error in GOFER-Combined is largely due to missed or low-quality observations by GOES-East (Table C3).

Based on the distribution of shortest distances from the GOFER to FRAP final perimeters, we estimate the spatial errors of GOFER-Combined as $0.75 \pm 0.21$ km for the mean and $2.86 \pm 1.14$ km for the maximum along its perimeter edges (Fig. C3). The spatial errors of GOFER-West are comparable with a mean of $0.87 \pm 0.31$ km and a maximum of $2.94 \pm 1.04$ km, while those of GOFER-East are higher with a mean of $1.44 \pm 0.44$ km and a maximum of $5.08 \pm 1.8$ km. The coarse resolution and geolocation errors of GOES affect

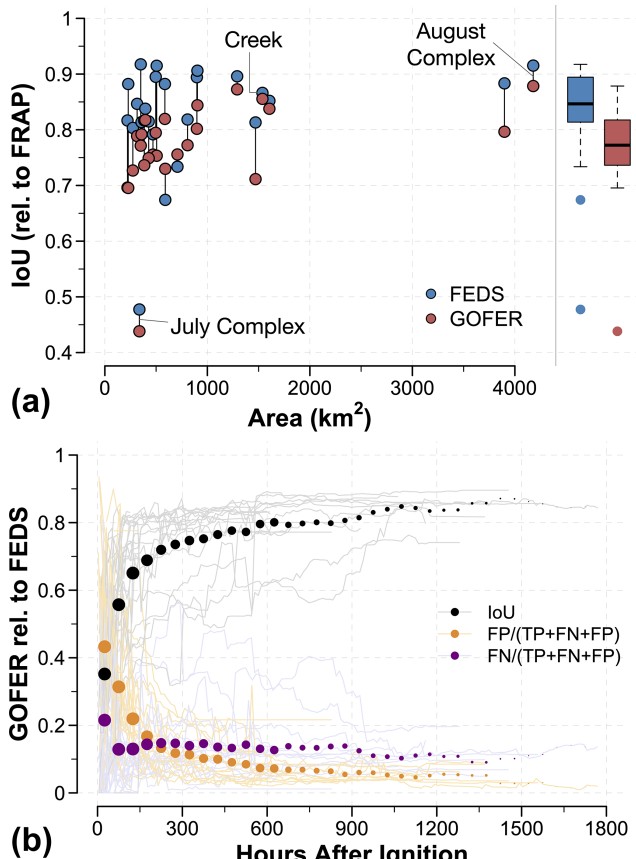

**(a)**

**(b)**

**Figure 8.** Spatial accuracy of GOFER perimeters compared to FRAP and FEDS. **(a)** The IoU of GOFER and FEDS final perimeters compared to FRAP for large fires in 2019–2021. The vertical lines connect GOFER and FEDS IoU for the same fires. **(b)** Accuracy metrics for evaluating GOFER against FEDS perimeters at 12 h intervals for large fires in 2019–2021. Along with the IoU, we show the fraction of false positives (FPs) and false negatives (FNs) within the union of the GOFER and FEDS perimeters (TP + FN + FP). In equivalent terms, the IoU is the fraction of true positives (TPs) within the union. Lines show the accuracy metrics for individual fires, dots show the average of all fires in 50 h bins, and the size of the dots represents the number of fires in each 50 h bin. Fires that straddled the border between CA and a neighboring state (i.e., Slater and Devil, Tamarack, and W-5 Cold Springs) were excluded, as FEDS perimeters cut off at the CA border.

the overall error along perimeter edges, while missing fire detections can cause large maximum errors, such as for the July Complex Fire.

## 3.2 The fire diurnal cycle derived from GOFER

The fire diurnal cycle is commonly derived from the FRP associated with active fires (Li et al., 2022; Giglio, 2007; Andela et al., 2015; Mu et al., 2011; Wiggins et al., 2020). Here, we instead track the fire diurnal cycle as the growth in fire-wide area, which the GOFER algorithm makes pos-

sible by resolving fire expansion at hourly intervals. Traditionally, burned-area products, available at daily to monthly timescales due to algorithm constraints, have lower temporal precision and frequency than needed to resolve diurnal variation (Giglio et al., 2018). As the fire front progresses, we expect the diurnal cycle of the fire-wide growth in area to coincide with or even precede that of active fires and FRP. This is because of a lingering fuel load behind the fire front that takes more time to fully burn through, resulting in active-fire detections inside the fire perimeter. Maxima in the diurnal cycle occur when the weather is hot, dry, and windy, such as in the afternoon, allowing the fire to easily burn through nearby dry fuels. Minima tend to occur when the weather is cool, wet, and stagnant, such as at night, when nearby fuels are too moist to catch on fire, thereby preventing fire spread (Balch et al., 2022).

We derive the fire diurnal cycle from the hourly fire-wide growth in the area of the GOFER perimeters. For GOFER-Combined, we observe two peaks in fire perimeter expansion during the afternoon (14:00–15:00 PDT) and evening (19:00–20:00 PDT), whereas GOFER-East and GOFER-West yield a single peak in growth during the afternoon (14:00–16:00 PDT) (Fig. 9a). The diurnal cycle of fire growth in GOFER-Combined closely mirrors that of GOES FRP (Wiggins et al., 2020). During the afternoon to evening hours (13:00–22:00 PDT), GOES-East, when compared to GOES-West, has higher peak-to-valley differences in FRP (−66 % versus −27 %) and fire detection confidence (−18 % versus −8 %), with noticeable minimums occurring during the day-to-night transition period (16:00–20:00 PDT) (Fig. 9b); similarly, the GOES-East active-fire pixel count deviates from that of GOES-West by −7 % on average during the same hours CE2. Because GOES-East observes California toward the edge of its disk view (Fig. B1), high solar zenith angles, sun glint issues, and mountainous terrain may explain the missed fire detections and lower fire detection confidence (Li et al., 2022). There may also be a positive nighttime fire detection bias, as smaller, cooler thermal anomalies are more easily distinguishable relative to the cooler background. Because the GOES-East and GOES-West fire detection confidence values are averaged, missed GOES-East detections can lead to false negatives or to burned area that is excluded from the perimeter. An example of this issue is seen on the southeastern edge of the Creek Fire, where the lack of GOES-East active fires led to an unburned inlet carved into the GOFER-Combined perimeter (Fig. C1).

As GOES-East is closer to the edge of its full-disk view than GOES-West in California, GOES-East observations are inherently less reliable and more subject to issues such as sun glint and viewing zenith angles. As such, the lower reliability of GOES-East during the day-to-night transition period likely drives the temporal artifacts in the fire diurnal cycle in GOFER-Combined. As GOFER-Combined uses a higher confidence threshold, the algorithm is more sensitive to missed or low-confidence detections, and the fire

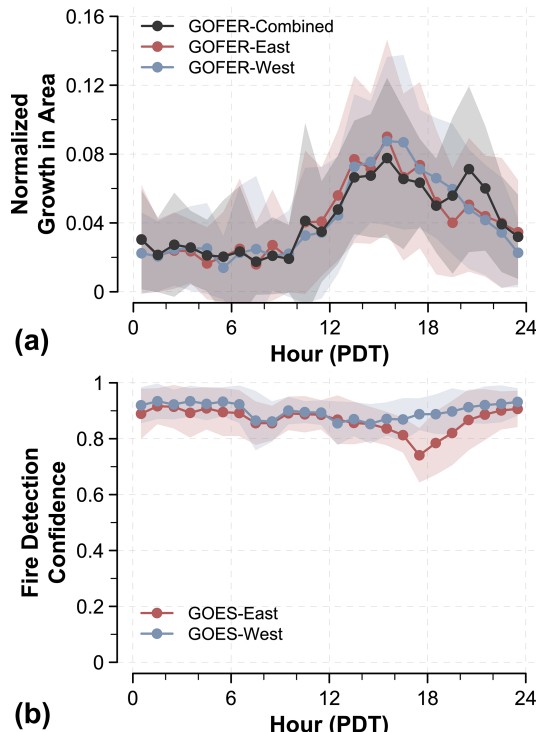

**Figure 9.** Average GOFER-derived fire diurnal cycle for 28 large CA wildfires from 2019 to 2021. The diurnal cycle is shown as the normalized hourly fraction of the **(a)** growth in area derived from the GOFER progression perimeters and **(b)** GOES fire detection confidence. For panel **(b)**, the spatial average is calculated from the maximum fire detection confidence of each pixel for each hour. The shaded areas represent $\pm 1$ standard deviation. CE3

growth in those late-afternoon hours will then be misallocated to evening hours. Even though GOFER-East relies only on GOES-East observations, its optimized confidence threshold (0.76) is less stringent than GOFER-Combined (0.95).
Low-confidence active pixels are less likely to be rejected in forming the hourly perimeter, thus resulting in more realistic diurnal cycles of fire growth in GOFER-East and GOFER-West compared with GOFER-Combined. This is a main limitation and area of future work for the GOFER-Combined al-
gorithm, as corrections are needed to boost the fire detection confidence during the day-to-night transition and assign different weights to GOES-East and GOES-West observations.

## 3.3 Assessing the GOFER active-fire lines and fire spread rates

For GOFER-Combined, the maximum $fline_{c=0.05}$ lengths range from 40 to 264 km, while $fline_r$ lengths range from 19 to 114 km (Table 1).

Figure 10 shows the time series of the GOFER-Combined $fline_c$ and $fline_r$ active-fire lengths and FEDSv2 active-fire
lengths for the Creek Fire. Both $fline_c$ and $fline_r$ lengths peak soon after ignition and gradually decrease as the fire expan-

sion slows down (Fig. 10a and b). Based on the correlation coefficient and slope, the FEDSv2 active-fire-line lengths are the closest to $fline_{c=0.05}$ ($r = 0.5$, $m = 0.68$, $p < 0.05$) at 12 h intervals (Fig. 10a). The calculation of $fline_c$ in the FEDS al-
25 gorithm is slightly different from the GOFER method, as the FEDS input active-fire data are represented as points rather than images of the fire detection confidence. To directly compare the two products, we use the same method to derive the 12 h aggregate $fline_r$ from FEDS and GOFER perimeters. For
the Creek Fire, the $fline_r$ lengths are moderately correlated ($r = 0.41$, $p < 0.05$) with an RMSE of 24.2 km (Fig. 10c). GOFER tends to underestimate the $fline_r$ length, with more values of zero, suggesting that some areas of expansion are not as well captured. This is partly because GOFER perime-
ters are less sinuous due to the lower spatial resolution of GOES and the neighborhood smoothing applied in the algorithm (Table C3). Slight day-to-day differences in the retrieval times of VIIRS fire detections also affect the comparison between GOFER and FEDS active-fire lines. While
GOFER uses all 10 min full-disk GOES images within each hour, VIIRS can only observe the state of the fire at its retrieval time, so the spatial extent and state of fire may have changed substantially at the end of the hour when GOFER and FEDS are compared.
For the 25 large CA fires, excluding cross-border fires, the overall correlation coefficient and slope between $fline_c$ and FEDS active-fire-line lengths decrease as the confidence threshold increases, while the RMSE increases
(Fig. C4). As such, the $fline_{c=0.05}$ should be considered as the default $fline_c$, with the $fline_c$ at higher confidence thresholds representing areas with increased likelihood of fire perimeter expansion. Relative to FEDS, GOFER-East $fline_r$ and $fline_c$ have consistently lower ac-
curacy than GOFER-Combined and GOFER-West. For GOFER-Combined, the $fline_{c=0.05}$ has an average $r = 0.49 \pm 0.25$, slope $= 0.48 \pm 0.24$, and RMSE $= 21 \pm 11$ km; the 12 h aggregate $fline_r$ has an average $r = 0.65 \pm 0.18$, slope $= 0.59 \pm 0.22$, and RMSE $= 17 \pm 10$ km (Fig. C4).
Figure 11 shows the time series of fire spread rates calculated using two different methods ($fspread_{MAE}$ and $fspread_{AWE}$) for the Creek Fire. The hourly $fspread_{MAE}$ and $fspread_{AWE}$ are strongly correlated ($r = 0.88$, $p < 0.05$), with $fspread_{MAE}$ being 2.59 times as high as $fspread_{AWE}$. For all 28 large CA fires, we find strong correlations of
$r = 0.93 \pm 0.05$ for GOFER-Combined, of $r = 0.94 \pm 0.02$ for GOFER-East, and of $r = 0.95 \pm 0.02$ for GOFER-West (Table C4). The ratio of $fspread_{MAE}$ to $fspread_{AWE}$ is $2.74 \pm 0.12$ for GOFER-Combined, $2.48 \pm 0.15$ for GOFER-
East, and $2.56 \pm 0.13$ for GOFER-West. For GOFER-Combined, the maximum $fspread_{MAE}$ ranges from 2.2 to $11.3 \ km\,h^{-1}$, whereas $fspread_{AWE}$ ranges from 0.9 to 10.8 km (Table 1). In rare cases, usually early in the fire's lifetime, we find higher $fspread_{AWE}$ than $fspread_{MAE}$ (e.g., Zogg Fire).
This happens when the fire grows explosively from a small perimeter and active-fire line in the previous time step.

**(a)** concurrent active fire line length

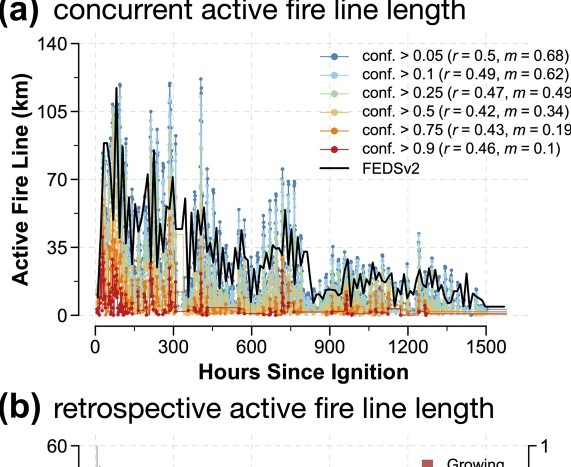

**(b)** retrospective active fire line length

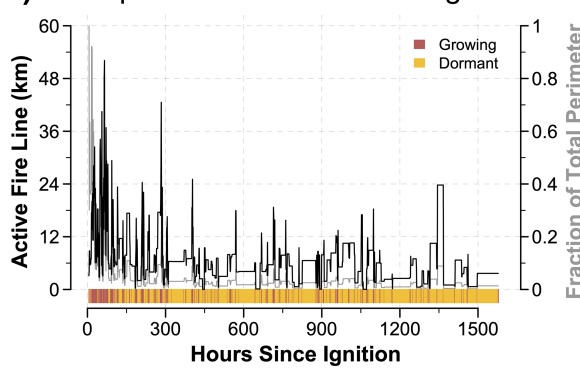

**(c)** 12 h retrospective active fire line length

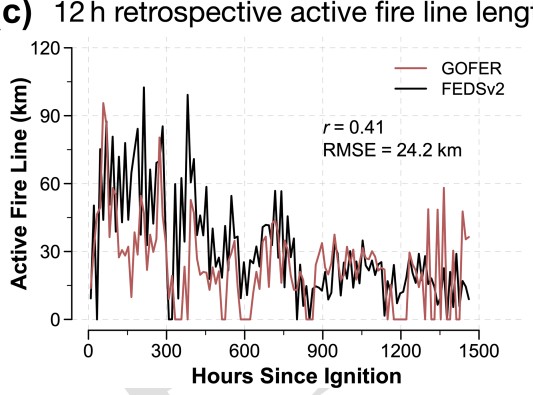

**Figure 10.** A comparison of GOFER-Combined and FEDS concurrent and retrospective active-fire-line lengths for the 2020 Creek Fire. **(a)** Concurrent active-fire lengths derived from GOFER perimeters and concurrent active-fire detections are shown using colored lines for different fire detection confidence cutoffs (0.05, 0.1, 0.25, 0.75, and 0.9). The 12 h FEDSv2 active-fire-line lengths are depicted by the black line. The correlation coefficient ($r$) and slope ($m$) between GOFER and FEDSv2 active-fire-line lengths are shown in the inset. **(b)** Retrospective active-fire-line lengths derived from GOFER perimeters are depicted by the black line. The fraction of the active-fire-line length with respect to the total perimeter length is depicted by the gray line. The bottom bar shows when the perimeter is growing (red) or dormant (orange). **(c)** The 12 h aggregate retrospective active-fire-line lengths derived from GOFER (red line) and FEDSv2 (black line) perimeters. All correlations shown are statistically significant at $p < 0.05$.

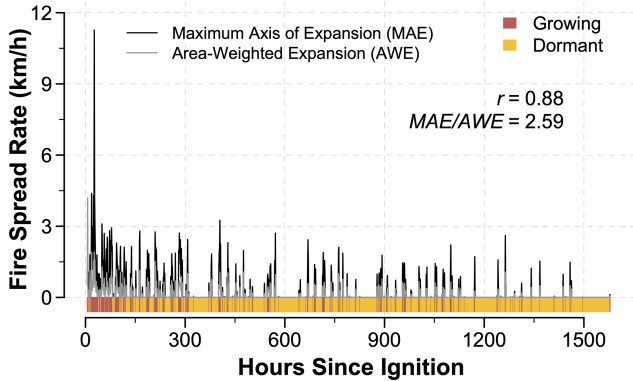

**Figure 11.** Hourly fire spread rate derived from GOFER-Combined perimeters for the 2020 Creek Fire. The fire spread rate is calculated using two methods: maximum axis of expansion (MAE, black line) and area-weighted expansion (AWE). The correlation coefficient ($r$) values between the MAE and AWE fire spread rates and the MAE / AWE ratio are shown inset. The bottom bar shows when the perimeter is growing (red) or dormant (orange).

## 3.4 Limitations, future work, and potential applications

### 3.4.1 Limitations

Here, we use 28 large fires in California from 2019 to 2021 to test the potential of the GOFER algorithm to track the hourly progression of large wildfires using 2 km GOES active-fire detections. While GOFER fills in temporal gaps in tracking fire progression, there are inherent limitations arising from the low spatial resolution of GOES observations, missed active-fire detections, and potential geolocation errors in the perimeters and the active-fire lines. In particular, GOFER is less reliable around waterbodies and mountainous terrain. While GOES-East and GOES-West observations can be combined to increase the overall spatial accuracy of GOES-derived perimeters, we find that, in California, GOFER-West is comparable to GOFER-Combined, and the use of GOES-East observations can detract from the spatial and temporal accuracy of GOFER-Combined. We expect that the suitability of the GOFER product for scientific applications, such as improving the fire diurnal cycle in emissions estimates or understanding the controls on extreme fire behavior, will grow as the algorithm is refined and additional fires are processed. However, GOFER cannot be used to understand fine-scale physical fire behavior, such as spotting or convection along the fire line, due to unnatural textures arising from the spatial limitations of GOES. Importantly, lessons learned in developing the GOFER algorithm may be applied to observations from future geostationary satellites over North and South America, such as NOAA's planned GeoXO (Geostationary Extended Observations) satellite system in the 2030s to replace the current GOES-R series with higher spatial resolution and additional bands (Adkins, 2022), and existing geostationary satellites over other regions, such as Hi-

mawari over East Asia, equatorial Asia, and Australia and Meteosat over Europe and Africa (Hally et al., 2016; Roberts and Wooster, 2008).

### 3.4.2 Future work and development

A useful direction for future work would be to apply the GOFER algorithm to a diverse sample of large fires across the GOES domain and test how its performance varies using observations from one or both satellites. Ground truth data for other regions may include perimeters provided by state and federal agencies or high-resolution burned-area mapping from Landsat and Sentinel. How GOFER-East and GOFER-West perform relative to each other depends largely on the longitudinal location of a given fire relative to the longitudinal position of the GOES satellites. We show that GOFER-West (IoU = 0.75) outperforms GOFER-East (IoU = 0.67) in mapping California fires, but we can also hypothesize that the reverse is true for fires in the Amazon and other biomes in South America. The spatial accuracy of mapping perimeters is influenced by substantial heterogeneity in the magnitude of parallax displacement and GOES pixel resolution across the GOES domain (Figs. B1 and B2). Thus, we can expect higher mapping accuracy for fires located at the center of the disk, near the Equator, and/or at low elevation than those at the edge of the disk, far from the Equator, and/or at high elevation.

As the GOFER algorithm is applied to fires outside California, small adjustments may include further optimizing tunable parameters. We currently implement a dynamic smoothing kernel size, the optimization for the confidence threshold and parallax adjustment factor, and early-perimeter adjustment. First, the smoothing kernel, which applies a neighborhood mean, removes the blockiness of the fire perimeter polygon that conforms to the pixelated footprint of the fire confidence image. Dynamically setting the smoothing kernel size equal to the GOES spatial resolution at a fire's location eliminates this blockiness and provides a universal method to calculate the smoothing kernel size for fires across the GOES domain. However, this smoothing induces errors in some fire perimeters that hug the coast, contain unburned islands (e.g., waterbodies), or encompass narrow swaths of burned area. To address these limitations, rules can be implemented for how the smoothing kernel is applied, such as according to nearby land cover. Second, we optimize the confidence threshold and parallax adjustment factor based on the IoU of GOFER and MTBS of the 10 largest fires in California in 2020. The two parameters can be tuned per fire, but this may lead to overtuning and substantially increases computation time. Additional optimization metrics may be considered, such as the maximum distance between true-positive and either false-positive or false-negative pixels, used by Google's current wildfire tracking system based on machine learning methods (Ben-Haim and Nevo, 2023). The kernel size for smoothing the fire detection confidence,

and the shape of the kernel itself, can also be tested as an additional tunable parameter. Future development of GOFER should consider how the optimal set of parameters differs by region and land cover by tuning the parameters on subsets of fires. Third, we apply early-perimeter adjustment to anchor the first perimeter close to or at the first GOES active-fire detection. The early-perimeter adjustment works by increasing the fire confidence if the maximum value is between 0.1 and 1. This adjustment targets fires that smolder for a long time before rapidly expanding, where the confidence of the GOES detections does not meet the threshold to create a perimeter. Additionally, as the footprint of a fire early in its lifetime (< 50 h after ignition) often encompasses only one to a few GOES active-fire pixels, the spatial accuracy of the GOFER early perimeters is low compared with FEDS (IoU < 0.5). One potential adjustment is shrinking each perimeter by its scaling factor (i.e., if < 1) to prevent overly inflating early perimeters derived from low-confidence detections. This process can then be tested on time steps with a maximum confidence below the minimum threshold of 0.1, which currently yield no perimeters; if successful, this adjustment will anchor the first perimeter of every fire to the time step with the first GOES active-fire detection.

Additional potential development areas include the adjustment of the fire detection confidence and the automation of the GOFER algorithm for use in near-real time. First, in GOFER, we currently convert the GOES fire mask codes to fire detection confidence following Restif and Hoffman (2020). While we optimize the confidence threshold against reference MTBS perimeters in the GOFER algorithm, resulting in spatial accuracy comparable to FEDS, we note that the initial remapping of fire mask codes includes user-specified elements, such as assigning a lower confidence value to saturated pixels versus processed pixels. In future work, it may be possible to use the detection confidence in 1 km MODIS active fires (MCD14ML), which ranges from 0 to 100, and the 375 m VIIRS active fires (VNP14IMGML), which consists of low-, medium-, and high-confidence categories, to readjust the conversion of GOES fire mask codes to fire detection confidence. Alignment of the fire detection confidence across GOES, MODIS, and VIIRS also enables integration of MODIS and VIIRS observations within the GOFER workflow and may ultimately improve GOFER's spatial accuracy. Second, the GOFER algorithm is currently semiautomated and processes each fire separately, relying on manual updates to a metadata dictionary containing that fire's bounding box and start and end time. Here, we tested the GOFER algorithm on fires that burned over 50 000 acres (202 km$^2$), but the lower size limit of fires that GOFER can map effectively should be explored. For operational, near-real-time use, GOFER needs to be able to identify individual fire events and determine these constraints automatically.

Finally, we rely on FEDS to evaluate the active-fire lines and fire spread rates here, both of which rely on the accuracy of the perimeters. More extensive evaluation and vali-

dation can be performed using aerial data and ground measurements. For example, future development of the concurrent active-fire lines in GOFER could use FRP to threshold and segment active-fire lines into fire intensity classes; however, this approach must account for uncertainties in the FRP calculated for saturated and low-quality fire pixels. To compare more directly to spread rates measured on the ground, the GOFER fire spread rates could be calculated for specific points or each grid cell in a predefined grid with the fspread$_\mathrm{MAE}$ approach.

### 3.4.3    Potential applications

We foresee several extensions and applications of the GOFER algorithm and product. First, GOFER can be used to improve the fire diurnal cycle for atmospheric modeling of smoke emissions. In current global fire emissions databases, the diurnal cycle is broadly generalized by land cover and is generally static from day to day throughout a fire's lifetime; for example, the 3 h fire diurnal cycles in the Global Fire Emissions Database (GFED) are derived from historical GOES observations from 2007 to 2009 and implemented as climatological means based on three land cover types (van der Werf et al., 2017; Mu et al., 2011). As is evident from GOFER, however, large fires may have explosive days of growth where burning extends from the afternoon to evening. In contrast, other days with slower fire spread are generally marked only by growth during the afternoon peak. Recently, GOES observations have been merged with VIIRS observations to estimate hourly fire emissions at 3 km spatial resolution in a top-down, FRP-based approach for the Regional ABI and VIIRS fire Emissions (RAVE) product (Li et al., 2022). Similarly, for a bottom-up, burned-area-based approach, the GOFER diurnal cycle of the fire-wide growth in area can be used to downscale the perimeters of select fires in existing fire progression products, such as FEDS, to hourly intervals. Second, the GOFER product can be used to build statistical and machine learning models to understand how temporal variations in weather, topography, fuels, and active-fire suppression at the active-fire line drive fire spread rate and fire-wide growth in area at an hourly scale. Owing to limitations in the spatial resolution in both the input and output data, GOFER is most suitable for 1D time series models. For example, the GOFER product can be used to explore periods of critical stress on firefighting resources, such as in mid-August and early September of 2020 when eight or nine large fires were simultaneously active (Fig. A1). Using the set of available fires in GOFER as case studies, we can identify periods when large fires are explosive or quiescent, including extreme cases when nighttime "brakes" on fire spread fail (Balch et al., 2022), causing evacuations and damaging structures. For spatial analyses, GOFER could be used as a secondary product to FEDS and high-resolution perimeters from state and federal agencies. GOFER and FEDS can be used to improve the parameterization of 3D fire spread mod-

els, such as ELMFIRE and WRF-Fire, during periods of extreme fire spread and active nighttime burning, which are often poorly estimated compared with satellite and aircraft observations (Stephens et al., 2022; Turney et al., 2023). The high temporal resolution of GOFER may enable advances in the initialization of the actively burning fire line in prognostic fire spread models (Stephens et al., 2022; Turney et al., 2023); however, potential geolocation errors should be accounted for. This could be done, for example, by perturbing the location and length of active-fire-line segments using an ensemble approach, with the sampling drawing upon the distribution of errors relative to reference perimeters.

## 4    Data availability

The GOFER product of the 28 fires in California from 2019 to 2021 is available on Zenodo at https://doi.org/10.5281/zenodo.8327264 (Liu et al., 2023). An online data visualization app for the GOFER product is available at https://globalfires.earthengine.app/view/gofer (Liu, 2024).

## 5    Code availability

The code for the GOFER algorithm is available at https://doi.org/10.5281/zenodo.8327264 (Liu et al., 2023).

## 6    Conclusion

In summary, we use GOES observations to develop the GOFER algorithm for deriving the hourly fire progression perimeters, active-fire lines, and fire spread rates of large wildfires. We test the algorithm using 28 fires that burned over 50 000 acres (202 km$^2$) in California from 2019 to 2021. We implement a parallax terrain correction with optimizations for the parallax adjustment factor and confidence threshold, early-perimeter adjustment, and a dynamic kernel for neighborhood smoothing. Relative to reference perimeters provided by FRAP, the spatial accuracy of GOFER (IoU = 0.77) is reasonable compared to the VIIRS-derived FEDSv2 (IoU = 0.83) at a 375 m spatial resolution. We apply two different methods to map active-fire lines (concurrent and retrospective) and calculate fire spread rates (MAE and AWE). GOFER resolves the time dimension of fire progression mapping to hourly intervals and can identify critical, explosive periods of fire spread. Opportunities for future development of the GOFER algorithm include resolving the day-to-night transition issues that skew the fire diurnal cycle of the fire-wide growth in area and testing GOFER in different ecosystems and regions across the GOES domain. Additionally, our GOFER product for the 28 large wildfires in California from 2019 to 2021 is a useful case study reference for modeling weather–human–fire relationships and improving estimates of fire emissions and smoke pollution.

## Appendix A: Study area: large wildfires in California

**Table A1.** Metadata for the 28 large wildfires in California from 2019 to 2021 that burned over 50 000 acres (202 km$^2$). Statistics are from the annual CAL FIRE *Redbooks*, which provide detailed information on each fire. The coordinates (longitude and latitude) and ignition times are from CAL FIRE and InciWeb; some ignition times are adjusted earlier if there are preceding GOES active-fire detections.

| No. | Fire name | Year | Area (acres) | Area (km$^2$) | Long | Lat | Ignition | (UTC) |
|---|---|---|---|---|---|---|---|---|
| 1 | Kincade | 2019 | 77 758 | 315 | −122.78 | 38.79 | 24 Oct 2019 | 04:00 |
| 2 | Walker | | 54 608 | 221 | −120.68 | 40.06 | 4 Sep 2019 | 21:00 |
| 3 | August Complex* | 2020 | 1 032 648 | 4179 | −122.67 | 39.78 | 16 Aug 2020 | 21:00 |
| 4 | Bobcat* | | 115 997 | 469 | −117.87 | 34.24 | 6 Sep 2020 | 19:00 |
| 5 | Creek* | | 379 895 | 1537 | −119.26 | 37.19 | 5 Sep 2020 | 01:00 |
| 6 | CZU Lightning Complex | | 86 509 | 350 | −122.22 | 37.17 | 16 Aug 2020 | 15:00 |
| 7 | Dolan* | | 124 924 | 506 | −121.60 | 36.12 | 18 Aug 2020 | 18:00 |
| 8 | Glass | | 67 484 | 273 | −122.50 | 38.56 | 27 Sep 2020 | 10:00 |
| 9 | July Complex | | 83 261 | 337 | −121.48 | 41.70 | 22 Jul 2020 | 17:00 |
| 10 | LNU Lightning Complex* | | 363 220 | 1470 | −122.15 | 38.48 | 17 Aug 2020 | 13:00 |
| 11 | North Complex* | | 318 935 | 1291 | −120.93 | 40.09 | 17 Aug 2020 | 16:00 |
| 12 | Red Salmon Complex* | | 144 698 | 586 | −123.43 | 41.19 | 27 Jul 2020 | 18:00 |
| 13 | SCU Lightning Complex* | | 396 625 | 1605 | −121.30 | 37.44 | 16 Aug 2020 | 11:00 |
| 14 | Slater and Devil* | | 166 127 | 672 | −123.38 | 41.77 | 8 Sep 2020 | 13:00 |
| 15 | SQF Complex* | | 175 019 | 708 | −118.50 | 36.26 | 19 Aug 2020 | 14:00 |
| 16 | W-5 Cold Springs | | 84 817 | 343 | −120.28 | 41.03 | 18 Aug 2020 | 18:00 |
| 17 | Zogg | | 56 338 | 228 | −122.57 | 40.54 | 27 Sep 2020 | 21:00 |
| 18 | Antelope | 2021 | 145 632 | 589 | −121.93 | 41.50 | 1 Aug 2021 | 17:00 |
| 19 | Beckwourth Complex | | 105 670 | 428 | −120.37 | 39.88 | 30 Jun 2021 | 23:00 |
| 20 | Caldor | | 221 835 | 898 | −120.54 | 38.59 | 15 Aug 2021 | 01:00 |
| 21 | Dixie | | 963 309 | 3898 | −121.38 | 39.88 | 14 Jul 2021 | 00:00 |
| 22 | KNP Complex | | 88 307 | 357 | −118.81 | 36.57 | 10 Sep 2021 | 14:00 |
| 23 | McCash | | 94 962 | 384 | −123.40 | 41.56 | 1 Aug 2021 | 02:00 |
| 24 | McFarland | | 122 653 | 496 | −123.03 | 40.35 | 30 Jul 2021 | 01:00 |
| 25 | Monument | | 223 124 | 903 | −123.34 | 40.75 | 31 Jul 2021 | 01:00 |
| 26 | River Complex | | 199 359 | 807 | −123.06 | 41.39 | 30 Jul 2021 | 21:00 |
| 27 | Tamarack | | 68 637 | 278 | −119.86 | 38.63 | 4 Jul 2021 | 18:00 |
| 28 | Windy | | 97 528 | 395 | −118.63 | 36.05 | 10 Sep 2021 | 10:00 |

* Fires used in parameter optimization.

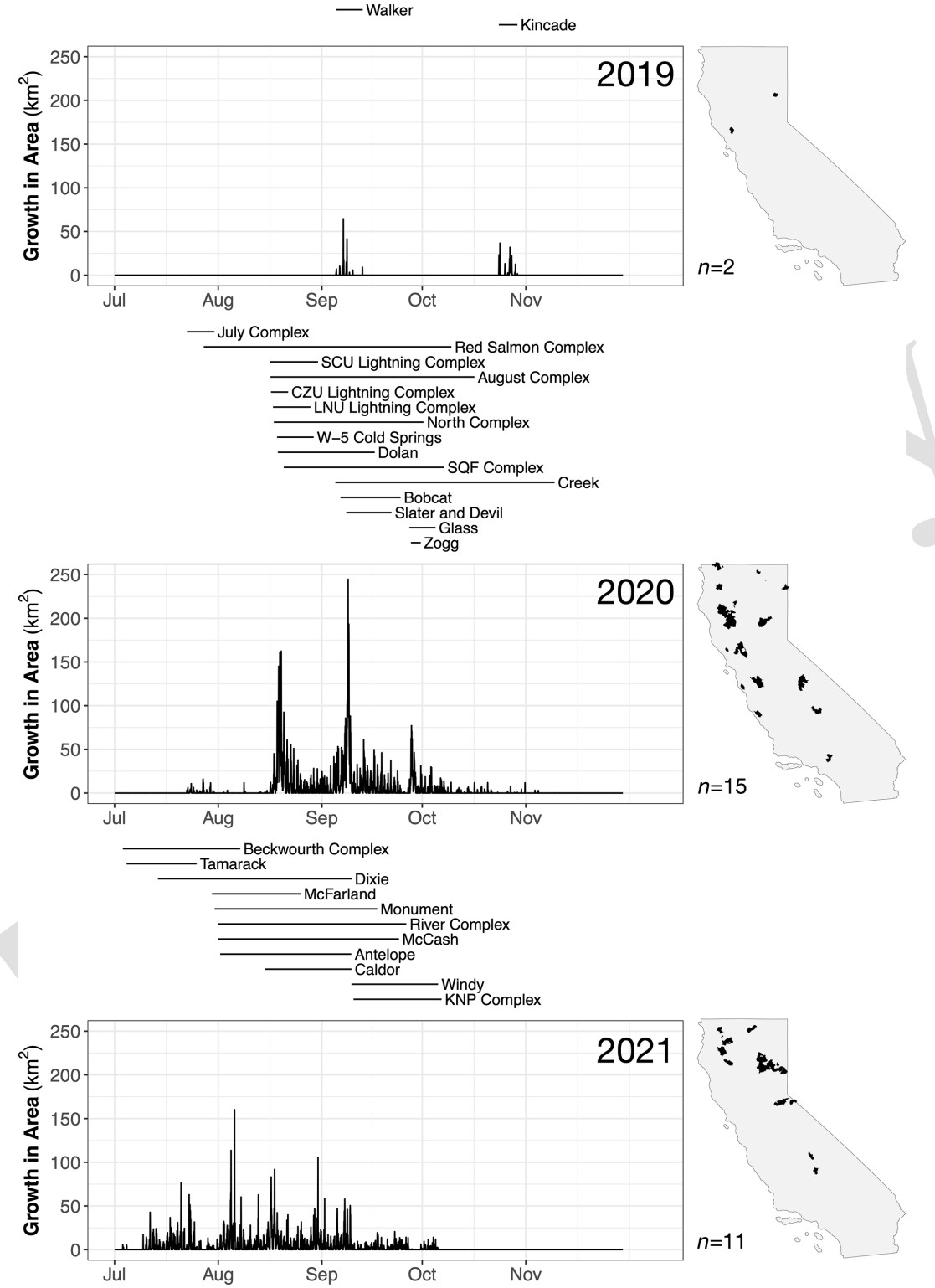

**Figure A1.** Time series of the hourly growth in area from 2019 to 2021 in the GOFER-Combined product. For each year, the growth in area (km$^2$) is summed across all fires. The horizontal lines above the time series represent the duration of active growth of each fire, ordered by start time. Annual maps of the locations of the fires are shown on the right.

## Appendix B: Development and optimization for GOES-based mapping of fire progression

**Table B1.** Remapping of GOES pixels by converting fire mask codes to continuous fire detection confidence values. All other pixels are presumed to be non-fire pixels and are assigned a fire detection confidence of zero.

| Fire mask category | | Fire detection |
| --- | --- | --- |
| Description | Code | confidence |
| Processed fire | 10 | 1 |
| Processed fire, filtered | 30 | |
| Saturated fire | 11 | 0.9 |
| Saturated fire, filtered | 31 | |
| Cloud contaminated fire | 12 | 0.8 |
| Cloud contaminated fire, filtered | 32 | |
| High probability fire | 13 | 0.5 |
| High probability fire, filtered | 33 | |
| Medium probability fire | 14 | 0.3 |
| Medium probability fire, filtered | 34 | |
| Low probability fire | 15 | 0.1 |
| Low probability fire, filtered | 35 | |

### Software details

*Input metadata dictionary.* For each fire, we set the spatial
and temporal constraints for processing GOES active fires
by examining the GOES active-fire time series and spatial
footprint. They are necessary to avoid computational time-
outs in GEE.

*Dynamic kernel.* The "reduce neighborhood" function to
smooth the fire detection confidence uses the boxcar opti-
mization, which is a fast method for computing the mean but
only works with square and rectangular kernels in GEE.

*Parallax correction.* To implement the GOES parallax cor-
rection in Earth Engine, we convert the Python code in the
"goes-ortho" package to JavaScript (Spestana et al., 2022). In
GEE, we use the "displace" function to correct the location
of GOES active-fire detections. We separately computed the
$x$ and $y$ components of the displacement for GOES-East and
GOES-West, in meters, between the coordinates (longitude
and latitude) of the DEM and satellite perspective as inputs
to this function. As a caveat, we must use a high-resolution or
downscaled DEM, as we find the displace function in GEE to
be inaccurate if the displacement is less than half the spatial
resolution of the DEM.

**Table B2.** The optimized confidence thresholds and parallax adjustment factors and smoothing kernel sizes used in GOFER.

| Version | Confidence threshold | Parallax adjustment factor | Smoothing kernel size (km) |
| --- | --- | --- | --- |
| GOFER-Combined | 0.95 | 0.85 | 1.6–1.7 |
| GOFER-East | 0.76 | 0.8 | 3.1–3.6 |
| GOFER-West | 0.83 | 1 | 2.5–2.7 |

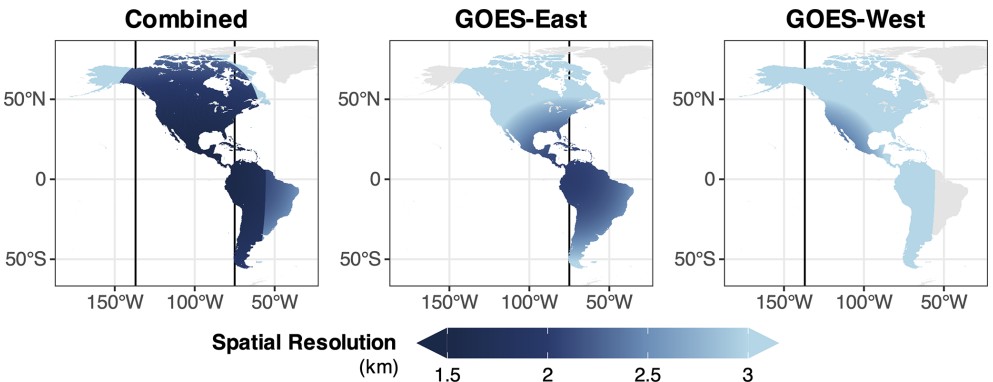

**Figure B1.** Spatial resolution of GOES-East, GOES-West, and combined GOES across the domain over land. The GOES spatial resolution, in kilometers, is calculated on the $0.25° \times 0.25°$ grid used by the Global Fire Emissions Database, version 4s (GFED4s). Vertical lines depict the longitudinal position of the GOES-East (75° W) and GOES-West (137° W) satellites.

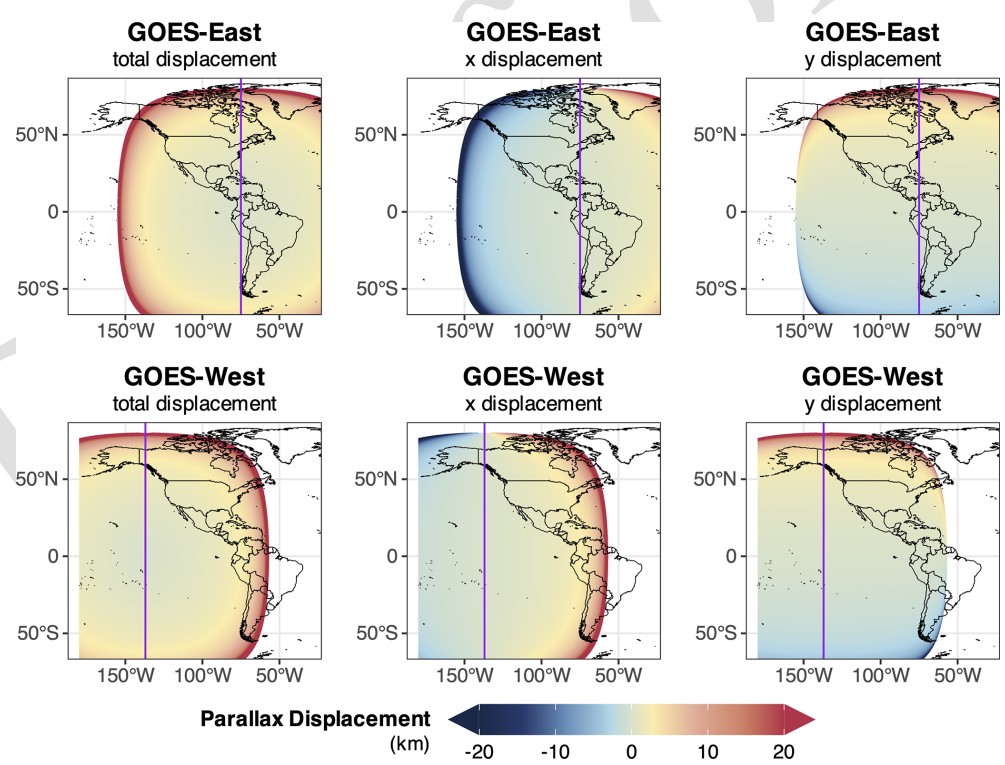

**Figure B2.** Parallax displacement in GOES-East and GOES-West images across the domain. The total, $x$ component, and $y$ component of the parallax displacement, in kilometers, are calculated for a hypothetical object at a 1 km elevation throughout the domain. For the $x$ component, negative values indicate that the object is displaced westward, whereas positive values that the object is displaced eastward. For the $y$ component, positive values indicate that the object is displaced northward, whereas negative values indicate that the object is displaced southward. The vertical purple lines depict the longitudinal position of the GOES-East (75° W) and GOES-West (137° W) satellites.

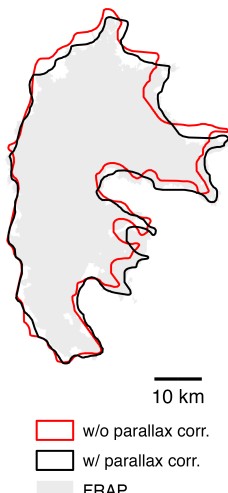

**Figure B3.** Effect of the parallax terrain correction on the final perimeter, using the Creek Fire as an example. The final perimeter of the Creek Fire with the parallax correction (red polygon) and without parallax correction for GOFER-Combined is shown alongside the FRAP perimeter (black polygon). For the uncorrected perimeter, we use a confidence threshold of 0.91, which yields the highest mean IoU among the 10 largest CA fires in 2020 when the parallax adjustment factor is zero (Fig. 4a).

**Table B3.** Tunable parameters in the GOFER algorithm.

| Tunable parameter | Definition and format |
|---|---|
| Fire mask codes to fire confidence conversion | Definition: converts the codes indicating the quality of active-fire detections to numeric values<br>Format: float, [0, 1] |
| Confidence threshold | Definition: delineates the border between burned and unburned area and indicates where to draw the fire perimeter<br>Format: float, [0, 1] |
| Smoothing kernel size | Definition: the radius of the kernel used to apply the neighborhood mean and smooth the GOES fire confidence<br>Format: float, > 0 |
| Parallax adjustment factor | Definition: the degree to which the parallax terrain adjustment is applied<br>Format: float, [0, 1] |
| Early-perimeter scaling | Definition: a scalar used to adjust the maximum fire confidence, relevant for time steps where the maximum value up to that time step falls below one<br>Format: float, [0, 1] |

## Appendix C: Evaluation and validation of the GOFER product

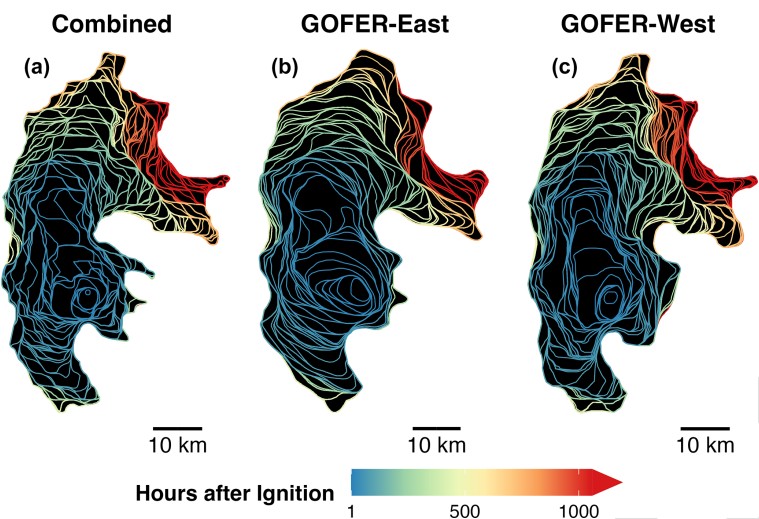

**Figure C1.** Spatiotemporal progression and comparison of the 2020 Creek Fire. Maps of the hourly GOFER progression for GOFER-Combined **(a)**, GOFER-East **(b)**, and GOFER-West **(c)**.

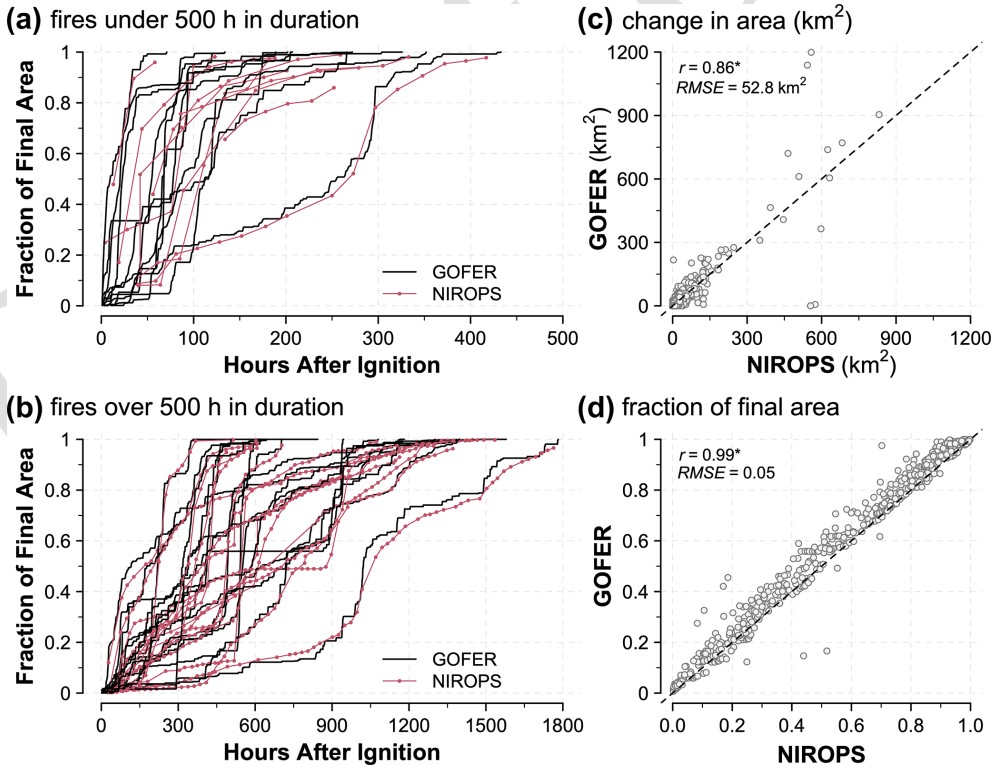

**Figure C2.** Comparison of the temporal progression of the 28 large fires in GOFER-Combined with NIROPS IR-based perimeters from NIFC. **(a, b)** Time series of the fraction of the final fire size for each fire from GOFER (black lines) and NIROPS (red lines) for fires **(a)** under 500 h in duration and **(b)** over 500 h in duration. For NIROPS, dots represent the availability of the IR imagery, which is almost all from nighttime flights. **(c, d)** Scatterplots of the **(c)** change in area between perimeter snapshots and **(d)** fractions of final fire size from GOFER and NIROPS for time steps when NIROPS perimeters are available. The correlation coefficient and RMSE are inset.

**Table C1.** The IoU calculated for GOFER-Combined, GOFER-East, GOFER-West, FEDSv2, and MTBS relative to FRAP.

| No. | Fire name | Year | IoU (GOFER, FRAP) | | | IoU (FEDS, FRAP) | IoU (MTBS, FRAP) |
|---|---|---|---|---|---|---|---|
| | | | GOFER-Combined | GOFER-East | GOFER-West | | |
| 1 | Kincade | 2019 | 0.79 | 0.72 | 0.76 | 0.85 | 0.99 |
| 2 | Walker | | 0.7 | 0.61 | 0.71 | 0.82 | 0.93 |
| 3 | August Complex[a] | 2020 | 0.88 | 0.83 | 0.87 | 0.92 | 0.95 |
| 4 | Bobcat[a] | | 0.76 | 0.63 | 0.68 | 0.79 | 0.98 |
| 5 | Creek[a] | | 0.86 | 0.77 | 0.83 | 0.87 | 0.98 |
| 6 | CZU Lightning Complex | | 0.77 | 0.73 | 0.87 | 0.92 | 0.97 |
| 7 | Dolan[a] | | 0.75 | 0.76 | 0.74 | 0.91 | 0.97 |
| 8 | Glass | | 0.73 | 0.62 | 0.69 | 0.8 | 0.99 |
| 9 | July Complex | | 0.44 | 0.52 | 0.49 | 0.48 | 0.97 |
| 10 | LNU Lightning Complex[a] | | 0.71 | 0.68 | 0.73 | 0.81 | 0.97 |
| 11 | North Complex[a] | | 0.87 | 0.73 | 0.87 | 0.9 | 0.98 |
| 12 | Red Salmon Complex[a] | | 0.82 | 0.73 | 0.83 | 0.88 | 0.97 |
| 13 | SCU Lightning Complex[a] | | 0.84 | 0.79 | 0.84 | 0.85 | 0.97 |
| 14 | Slater and Devil[a,b] | | 0.77 | 0.64 | 0.78 | – | 0.98 |
| 15 | SQF Complex[a] | | 0.76 | 0.66 | 0.71 | 0.73 | 0.96 |
| 16 | W-5 Cold Springs[b] | | 0.79 | 0.59 | 0.74 | – | 0.98 |
| 17 | Zogg | | 0.7 | 0.56 | 0.76 | 0.88 | 0.99 |
| 18 | Antelope | 2021 | 0.73 | 0.6 | 0.73 | 0.67 | 0.95 |
| 19 | Beckwourth Complex | | 0.75 | 0.53 | 0.71 | 0.81 | 0.96 |
| 20 | Caldor | | 0.8 | 0.71 | 0.8 | 0.89 | 0.97 |
| 21 | Dixie | | 0.8 | 0.68 | 0.78 | 0.88 | 0.97 |
| 22 | KNP Complex | | 0.79 | 0.67 | 0.76 | 0.81 | 0.98 |
| 23 | McCash | | 0.74 | 0.65 | 0.73 | 0.82 | 0.97 |
| 24 | McFarland | | 0.79 | 0.75 | 0.71 | 0.9 | 0.97 |
| 25 | Monument | | 0.84 | 0.76 | 0.84 | 0.91 | 0.98 |
| 26 | River Complex | | 0.77 | 0.65 | 0.74 | 0.82 | 0.92 |
| 27 | Tamarack[b] | | 0.69 | 0.53 | 0.63 | – | 0.95 |
| 28 | Windy | | 0.82 | 0.7 | 0.77 | 0.84 | 0.99 |
| | Mean IoU (all fires) | | $0.77 \pm 0.08$ | $0.67 \pm 0.08$ | $0.75 \pm 0.08$ | – | $0.97 \pm 0.02$ |
| | Mean IoU (excludes cross-border fires) | | $0.77 \pm 0.09$ | $0.68 \pm 0.08$ | $0.76 \pm 0.08$ | $0.83 \pm 0.09$ | $0.97 \pm 0.02$ |

[a] Fires used in parameter optimization. [b] The IoU for cross-border fires is omitted for FEDS, as the perimeter of these fires are not fully mapped.

**Table C2.** Comparison of the final perimeter sinuosity for the 25 non-cross-border fires. The sinuosity of the fire perimeter is defined as the length of the perimeter divided by the diameter of a circle with the same area.

| Source | Sinuosity ($\pm 1\,\mathrm{SD}$)* |
|---|---|
| GOFER-Combined | $4.9 \pm 1.2$ |
| GOFER-East | $4.3 \pm 0.8$ |
| GOFER-West | $4.6 \pm 0.9$ |
| FEDSv2 | $5.9 \pm 1.9$ |
| FRAP | $14.3 \pm 6.7$ |

* SD: standard deviation.

**Table C3.** The number of damaged and destroyed structures within GOFER-Combined, GOFER-East, and GOFER-West final perimeters. Undamaged and inaccessible structures are excluded.

| No. | Fire name | Year | Within perimeter (%) | | | Total ($n$) |
|-----|-----------|------|----------------|-----------|-----------|-------------|
| | | | GOFER-Combined | GOFER-East | GOFER-West | |
| 1 | Kincade | 2019 | 96 | 98 | 96 | 434 |
| 3 | August Complex* | 2020 | 95 | 98 | 95 | 99 |
| 4 | Bobcat* | | 99 | 100 | 100 | 216 |
| 5 | Creek* | | 96 | 99 | 100 | 929 |
| 6 | CZU Lightning Complex* | | 72 | 83 | 96 | 1630 |
| 8 | Glass | | 97 | 97 | 100 | 1810 |
| 10 | LNU Lightning Complex* | | 91 | 95 | 97 | 1723 |
| 11 | North Complex* | | 98 | 98 | 98 | 2471 |
| 13 | SCU Lightning Complex* | | 100 | 100 | 100 | 251 |
| 14 | Slater and Devil* | | 100 | 100 | 100 | 377 |
| 15 | SQF Complex* | | 100 | 100 | 100 | 244 |
| 17 | Zogg | | 100 | 100 | 100 | 231 |
| 18 | Antelope | 2021 | 100 | 100 | 100 | 24 |
| 19 | Beckwourth Complex | | 100 | 100 | 100 | 171 |
| 20 | Caldor | | 100 | 100 | 100 | 1086 |
| 21 | Dixie | | 90 | 91 | 99 | 1405 |
| 24 | McFarland | | 94 | 94 | 100 | 47 |
| 25 | Monument | | 97 | 100 | 97 | 30 |
| 27 | Tamarack | | 100 | 100 | 100 | 17 |
| 28 | Windy | | 10 | 10 | 100 | 21 |
| Mean | | | $92 \pm 20$ | $93 \pm 20$ | $99 \pm 2$ | |

* Fires used in parameter optimization.

**Table C4.** Comparison of GOFER fire spread rates derived from the MAE (maximum axis of expansion) and AWE (area-weighted expansion) methods.

| Version | Correlation coefficient ($r$, $\pm 1\,\text{SD}$)* | MAE / AWE ($\pm 1\,\text{SD}$)* |
|---------|-------------------------------------|-------------------|
| GOFER-Combined | $0.93 \pm 0.05$ | $2.74 \pm 0.12$ |
| GOFER-East | $0.94 \pm 0.02$ | $2.48 \pm 0.15$ |
| GOFER-West | $0.95 \pm 0.02$ | $2.56 \pm 0.13$ |

* SD: standard deviation.

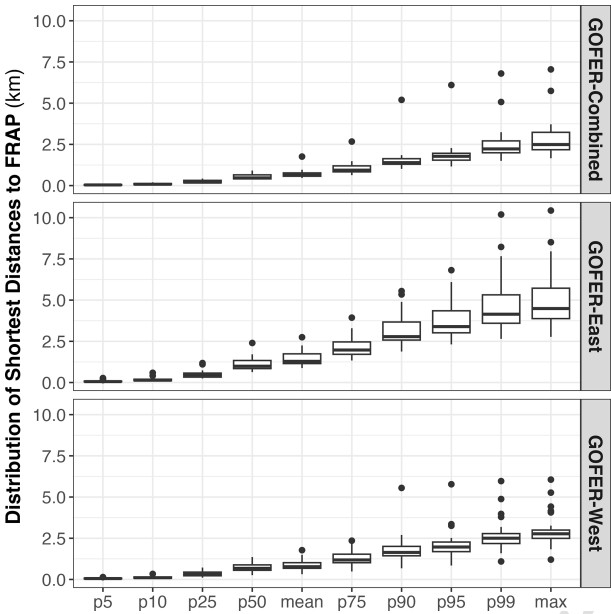

**Figure C3.** Distribution of shortest distances from GOFER to FRAP final perimeters for the 28 fires in this study. Each box plot represents the distribution of the shortest distances among the 28 fires at different breakpoints in the distribution for each fire: mean; median; and the 5th, 10th, 25th, 75th, 90th, 95th, and 99th percentiles. Separate analyses are shown for the GOFER-Combined, GOFER-East, and GOFER-West perimeters.

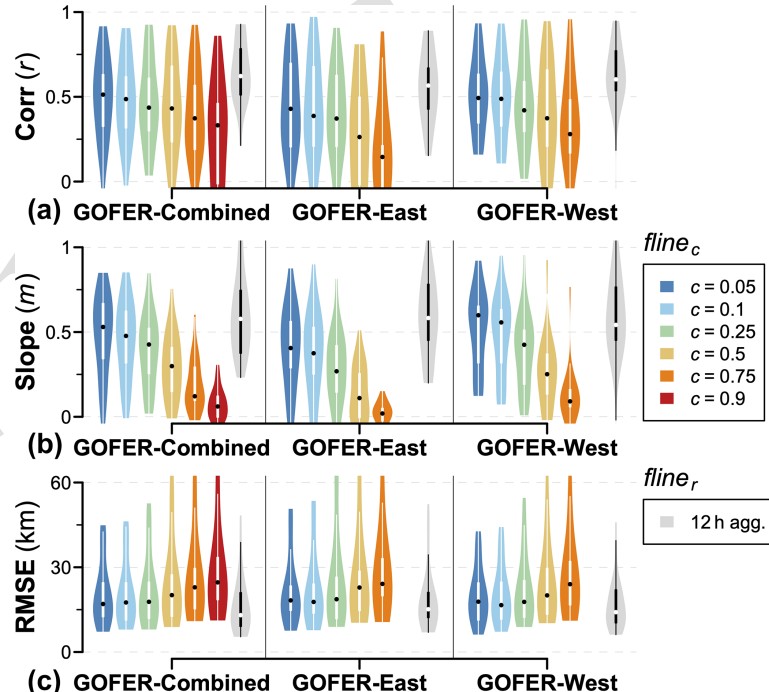

**Figure C4.** Comparison of GOFER and FEDSv2 active-fire-line lengths. The violin plots show the distribution of **(a)** correlation coefficients, **(b)** slopes, and **(c)** RMSEs for 25 non-cross-border CA fires from 2019 to 2021. GOFER $fline_c$ lengths are compared to the out-of-box FEDSv2 active-fire-line lengths, while the 12 h aggregate $fline_r$ lengths are calculated using the same method for GOFER and FEDSv2. $fline_{c=0.9}$ was not derived for GOFER-East nor GOFER-West, as the optimized confidence thresholds used to map perimeters were lower than 0.9.

**Author contributions.** TL and JTR designed the study. TL developed the code and carried out the data processing and analysis. All authors contributed to the interpretation of the results. TL prepared the manuscript with contributions from all co-authors.

**Competing interests.** The contact author has declared that none of the authors has any competing interests.

**Disclaimer.** Publisher's note: Copernicus Publications remains neutral with regard to jurisdictional claims made in the text, published maps, institutional affiliations, or any other geographical representation in this paper. While Copernicus Publications makes every effort to include appropriate place names, the final responsibility lies with the authors.

**Acknowledgements.** Tianjia Liu acknowledges support from the NOAA Climate and Global Change Postdoctoral Fellowship Program, administered by UCAR's Cooperative Programs for the Advancement of Earth System Science (CPAESS) under the NOAA Science Collaboration Program award no. NA21OAR4310383. James T. Randerson acknowledges support from the US DOE Office of Science RUBISCO Science Focus Area and NASA's Modeling Analysis and Prediction program (grant no. 80NSSC21K1362). Yang Chen and James T. Randerson acknowledge support from the US DOE LLNL-LDRD program (grant no. DE-AC52-07NA27344 and project no. 22-ERD-008, "Multiscale Wildfire Simulation Framework and Remote Sensing"). Efi Foufoula-Georgiou acknowledges support from NASA's Global Precipitation Measuring Mission and Weather and Atmospheric Dynamics (grant nos. 80NSSC22K0597 and 80NSSC23K1304) and from NSF (grant nos. IIS 2324008). Douglas C. Morton, James T. Randerson., Yang Chen, and Elizabeth B. Wiggins acknowledge support from NASA's Earth Information System (EIS) Fire project.

**Financial support.** This research has been supported by the University Corporation for Atmospheric Research (grant no. NA21OAR4310383), the National Aeronautics and Space Administration (grant nos. 80NSSC21K1362, 80NSSC22K0597, and 80NSSC23K1304 and the Earth Information System – EIS), the US Department of Energy (grant nos. DE-AC52-07NA27344 and 22-ERD-008 and the RUBISCO Science Focus Area), and the Directorate for Computer and Information Science and Engineering (grant no. 2324008).

**Review statement.** This paper was edited by Jia Yang and reviewed by two anonymous referees.

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

## Remarks from the language copy-editor