# Peer review of "Systematically tracking the hourly progression of large wildfires using GOES satellite observations"

_Earth System Science Data, 2023_

## Author Comment (AC1)

**Response to Referee #1**

The authors present the "GOFER" algorithm and derived data sets (GOFER-Combined, GOFER-West, GOFER-East) of hourly fire perimeters, active fire lines, and fire spread rates for 28 large wildfires in California from 2019-2021. Results are partially evaluated using a combination of fire perimeters from VIIRS data and final fire perimeters from California's Fire and Resource Assessment Program (FRAP).

We thank the reviewer for their helpful and constructive comments that have helped us improve our paper. Our point-by-point responses to these comments are listed below, and additional technical changes are listed at the end of our response. Our major changes include (1) restructuring the text in the methods and discussion sections for clarity, (2) adding an analysis of the spatial errors in GOFER perimeters relative to FRAP, and (3) adding a validation of intermediate GOFER perimeters with reference perimeters derived from aerial infrared imagery.

Specific Comments

* Section 1: Introduction

Line 59: "create a dataset" - Please choose another term than data/dataset and use throughout to make clear what are shown are not longer measurements, but results of calculations - perhaps some type of "product"?

We have revised "dataset" to "product" throughout the manuscript.

* Section 2.3 Using GOES active fire detections to derive hourly perimeters

Section 2.3.1: This section aims to provide a description of the GOFER algorithm but runs into some difficulty. The authors extend an approach originally described on a GEE blog (Restif and Hoffman, 2020) which here is cursorily described as a "GOES-based image-to-vector method to map fire perimeters in GEE". The remainder of the algorithm overview consists of describing optimizations/adjustments made to the original GEE approach clearly with the assumption that readers are already familiar with that approach. This is a mistake, I think, and more detail about the original method should be provided. The manuscript is cluttered with poorly defined and/or confusing quantities without this additional background material. An example is the "fire detection 'confidence' values" How were the arbitrary values in Table B1 selected? Why are saturated fire pixels assigned lower confidence? Table B3, which is deferred to the software (not algorithm) description, is of little help here, e.g., the fire confidence conversion "converts the codes indicating the quality of active fire detections to numeric values", but the fire detection codes are already numeric values.

Thank you for this suggestion. We have added a brief description of the method described by Restif and Hoffman (2020) at the beginning of Section 2.3.1:

L148-172: "Restif and Hoffman (2020) show a step-by-step example of a GOES-based image-to-vector method to map fire perimeters in GEE for the 2019 Kincade Fire in California. After filtering GOES-East and GOES-West observations over a 2-week period and over an AOI defined as a 40-km buffer of the point location of the Kincade Fire, the GOES "fire mask codes" provided by the Fire Detection and Characterization (FDC) algorithm are remapped to fire detection "confidence" values (Table B1). This remapping arbitrarily weights the fire pixels and non-fire pixels on a continuous, interpretable scale that ranges from 0-1. Based on threshold tests, the GOES FDC algorithm categorizes the quality of the fire pixels as "processed," "saturated," "cloud contaminated," "high probability," "medium probability," or "low probability" (Schmidt et al., 2012). "Processed" and "saturated" codes refer to the highest quality fire pixels, while "cloud contaminated," "high probability," "medium probability," and "low probability" codes refer to lower-quality fire pixels that may be false alarms. For each satellite, the maximum fire detection confidence is calculated from GOES images retrieved within the input temporal limits, and the GOES-East and GOES-West maximum fire detection values are combined by taking the minimum. Next, the combined GOES fire detection confidence map is smoothed using a 2-km square kernel. A confidence threshold of 0.6 is applied to mask low-confidence areas, and the image is then converted into a vector at a spatial resolution of 200 m. Since the resulting vector retains unnatural edges from the footprint of the image pixels, this artificial complexity in the vector is simplified within a maximum error of 500 m, thereby smoothing any jagged edges."

Here we expand and improve the Restif and Hoffman (2020) method by adding four optimizations or adjustments in our GOFER algorithm: (1) dynamic smoothing kernel size, (2) early perimeter adjustment, (3) parallax terrain correction, and (4) confidence threshold optimization. Specifically, we reduce the arbitrary selection of parameters by optimizing against perimeters derived from high-resolution satellite imagery, increase the geolocation accuracy of GOES fire pixels with a parallax terrain correction, and improve the mapping of early perimeters."

We further clarify in L191-192: "In step 1, following Restif and Hoffman (2020), we assign GOES-East and GOES-West fire mask codes as fire detection confidence values (Table B1)."

In the discussion section, we clarify that this is an area of future development:
L840-848: "First, in GOFER, we currently convert the GOES fire mask codes to fire detection confidence following Restif and Hoffman (2020). While we optimize the confidence threshold against reference MTBS perimeters in the GOFER algorithm, resulting in spatial accuracy comparable to FEDS, we note that the initial remapping of fire mask codes includes user-specified elements, such as assigning a lower confidence value to "saturated" pixels versus "processed" pixels. In future work, it may be possible to use the detection confidence in 1-km MODIS active fires (MCD14ML), which ranges from 0 to 100, and the 375-m VIIRS active fires (VNP14IMGML), which consists of "low," "medium," and "high" confidence categories, to readjust the conversion of GOES fire mask codes to fire detection confidence. Alignment of the fire detection confidence across GOES, MODIS, and VIIRS also enables integration of MODIS and VIIRS observations within the GOFER workflow and may ultimately improve GOFER's spatial accuracy."

The GOES-R overall percentage of saturated pixels is below 5% (Schmidt et al., 2020), so we do not expect the future adjustment of the confidence conversion value for saturated pixels to change the GOFER perimeters significantly.

Figure 2: This figure (upper right panel) implies that the authors assume there is zero geolocation error in the respective GOES ABI scans. This is not so, and the authors should include a brief discussion of the impact that these errors can have on their results. I believe they will find that this error is not much smaller than the downscaled 1.7 km resolution they claim to achieve by intersecting GOES-16 and GOES-17 pixels. A useful starting point might be the EMOSS team at NOAA (https://www.ospo.noaa.gov/Operations/GOES/goes-inrstats.html).

We have reviewed the GOES-R MRD document and added the following:
L108-110: "The nominal product mapping accuracy for the GOES-R Series Fire/Hot Spot Characterization product is 1 km (https://www.goes-r.gov/syseng/docs/MRD.pdf)."

To quantify the magnitude of the spatial errors of the GOFER perimeters, we have added an analysis of the distribution of shortest distances from the GOFER perimeter to the FRAP perimeter with additions in the methods, results/discussion, and Figure C3.

In Section 2.6, we add:
L409-415: "Further, to quantify the spatial error between the GOFER and FRAP final perimeters, we calculate "breakpoints" in the distribution of shortest distances from the GOFER perimeter to the FRAP perimeter. These breakpoints are defined by the mean and several percentiles, including the median and maximum, and the magnitude of these distances represents the deviation of the GOFER perimeter from the ground truth. This spatial error is induced by a combination of coarse spatial resolution, geolocation error, and missing fire detections in GOES. Our analysis is similar to the evaluation described in Ben-Haim and Nevo (2023) for GOES-derived fire perimeters but incorporates both false positives and false negatives in one metric."

In Section 3.1, we add:
L538-543: "Based on the distribution of shortest distances from the GOFER to FRAP final perimeters, we estimate the spatial errors of GOFER-Combined as $0.75 \pm 0.21$ km for the mean and $2.86 \pm 1.14$ km for the maximum along its perimeter edges (Figure C3). The spatial errors of GOFER-West are comparable with a mean of $0.87 \pm 0.31$ km and a maximum of $2.94 \pm 1.04$ km, while those of GOFER-East are higher with a mean of $1.44 \pm 0.44$ km and a maximum of $5.08 \pm 1.8$ km. The coarse resolution and geolocation errors of GOES affect the overall error along perimeter edges, while missing fire detections can cause large maximum errors, such as for the July Complex Fire."

[Figure]

**Figure C3: Distribution of shortest distances from GOFER to FRAP final perimeters for the 28 fires in this study.** Each boxplot represents the distribution of the shortest distances among the 28 fires at different breakpoints in the distribution for each fire: mean, median, and the 5th, 10th, 25th, 75th, 90th, 95th, and 99th percentiles. Separate analyses are shown for GOFER-Combined, GOFER-East, and GOFER-West perimeters.

We add a validation using high-resolution perimeters derived from airborne infrared imagery provided by the National Interagency Fire Center (NIFC). This analysis provides validation of intermediate perimeters, following the approach in Chen et al., (2022). Please see changes in Section 2.2, 2.6, 3.1, and Figure C2.

Section 2.2, L136-142: "NIFC provides high-resolution intermediate perimeters derived from airborne infrared (IR) imagery by trained analysts (https://data-nifc.opendata.arcgis.com/). The availability of these perimeters is sparse, varying from fire to fire and affected by cloud cover, thick smoke, and availability of flights and coverage area. For example, almost all flights are during nighttime, and some sections of the fire may not be mapped during a particular flight. We use the IR perimeters from the U.S. Forest Service National Infrared Operations (NIROPS) Unit. After filtering the NIROPS perimeters for data quality (e.g. missing metadata, small flight coverage) and matching with GOFER perimeters by the nearest hour, our reference dataset comprises over 650 snapshots across the 28 fires."

Section 2.6, L418-422: "To validate the temporal progression of GOFER perimeters, we use NIROPS perimeters derived from airborne IR imagery. We track the cumulative fire area accumulated relative to the final fire size. Because NIROPS perimeters are relatively sparse and almost all during nighttime, we additionally evaluate the performance of GOFER relative to FEDS over each fire's lifetime to check when the GOFER perimeters are relatively stable in spatial accuracy…"

Section 3.1, L475-483: "The overall temporal progression of the cumulative change in fire-wide area in GOFER agrees well with NIROPS. For example, for hours that NIROPS perimeters are available, we find a strong correlation between the change in fire area ($r = 0.86$, RMSE = 52.8 km$^2$) between NIROPS snapshots and fractions of final fire size ($r = 0.99$, RMSE = 0.05) from GOFER-Combined and NIROPS (Figure C2). High RMSE in the change in fire area mainly stems from a few instances of high bias between some snapshots in complex fires. The median absolute bias is 6.7 km$^2$, while the mean absolute bias is 16.7 km$^2$. GOFER has a median positive bias of 0.02 in the fractions of final fire size, suggesting that perimeter growth accumulates slightly earlier for GOFER than

for NIROPS. As a caveat, NIROPS does not fully map the fire for some snapshots, so some areas of active growth may be missing. The discrepancies may also indicate that GOFER is unable to pick up small increments of growth later in the fire's lifetime when fire front is less active."

[Figure]

**Figure C2: Comparison of the temporal progression of the 28 large fires in GOFER-Combined with NIROPS IR-based perimeters from NIFC. (a-b)** Timeseries of the fraction of the final fire size for each fire from GOFER (black lines) and (red lines) for fires **(a)** over 500 h in duration and **(b)** under 500 h in duration. For NIROPS, dots represent the availability of the IR imagery, which are almost all from nighttime flights. **(c-d)** Scatterplots of the **(c)** change in area between perimeter snapshots **(d)** fractions of final fire size from GOFER and NIROPS for timesteps when NIROPS perimeters are available. The correlation coefficient and RMSE are inset.

Section 2.3.2 "we create a dictionary of input values": Presumably "dictionary" (as opposed to, e.g., list or table) refers to the Python data structure. Clarify please.

We have clarified in Section 2.3.2, L240-242 that "dictionary" refers to the data structure often used in coding: "Here, "dictionary" refers to the data structure in code stored as "key" and "value" pairs, where the keys, or user-specified words, are used to retrieve the corresponding values."

Section 2.3.3.3: Please clarify if the "10 largest fires in California in 2020" are among those fires in the final 28-fire GOFER data set. Here and elsewhere the manuscript is unclear about which reference fires were used for parameter optimization and which reference fires were used to evaluate the final GOFER database. Presumably, the authors are not using the same reference data for optimization and evaluation, but this needs to be made clear in the text.

We have clarified that the 10 largest fires are among those in the final 28 fires in L301-302: "the parameter search uses the 10 largest fires in California in 2020, a subset of the 28 fires in this study." In our revision, we label these 10 fires in Tables 1, A1, C1, and C3 with an asterisk.

We present the evaluation metrics for all 28 fires in the study. We note in Section 3.1, L508-510 that: "In addition, the average IoU for the 10 megafires in 2020 that we used to optimize parameters is similar to the IoU for the 7 megafires in 2021 (e.g., the IoU for GOFER-Combined and FRAP is 0.8 for 2020 fires and 0.78 for 2021 fires). The lack of a significant drop in IoU suggests that our parameters are not over-tuned to those 10 fires in 2020."

Figure 4: IoU first appears here but is not defined. I am sure Intersection of Union is intended, and in this case the authors should clarify if they are calculating the metric with the actual perimeters or rectangular bounding boxes as is often the case in ML literature.

We have spelled out IoU as Intersection-over-Union (IoU) in the Figure 4 caption. In Section 2.3.3.3, L298-301 we revised the IoU definition and clarified that the perimeters are used to calculate the IoU: "The IoU, or Jaccard index, is a common metric for evaluating spatial accuracy against ground truth data in object detection. Here the IoU is calculated as the area of overlap over the area of union using the fire perimeters, in which an IoU of 0 indicates no agreement and IoU of 1 indicates perfect agreement."

* Section 2.4 Derived fire metrics

2.4.1 Active fire line

Line 247: "We output fline_c at hourly confidence thresholds c of 0.05, 0.1, 0.25, 0.5, 0.75, and 0.9" It is unclear if this means that six different fline_c estimates are provided in the GOFER data set for each time step. If so, which one should those using the GOFER data select? Slightly more detail (but not enough) is given in the first paragraph of Section 3.3, but this should be included with the definition.

For each timestep, the concurrent fire line geometry and length for all confidence thresholds are provided. We have moved the description in Section 3.3 to Section 2.4.1:

L329-336, 344-350: "For each hour, we separately output $fline_c$ at confidence thresholds $c$ of 0.05, 0.1, 0.25, 0.5, 0.75, and 0.9; this set of $fline_c$ at varying thresholds allows us to narrow down perimeter segments with the most intense burning progressively. A lax threshold, such as $fline_{c=0.05}$, uses most of the active fire detections during that hour, while a strict threshold, such as $fline_{c=0.9}$, only uses high confidence detections to create the hourly GOES fire perimeters. The $fline_{c=0.05}$ is most comparable to active fire lines in other satellite-derived products such as FEDS, which uses all active fire pixels intersecting with the perimeter. $fline_c$ with stricter thresholds correspond to areas with higher fire intensity. We convert the perimeters from polygons to linestrings and use a buffer of 100 m around the perimeter to extract intersecting active fire pixels with fire detection confidence above the defined threshold.

In general, the $fline_c$ can be calculated in near-real-time along with perimeters and is most useful for identifying potential areas of spread along the perimeter and testing predictive models of future fire growth. The set of $fline_c$ at different confidence thresholds can be used in tandem to identify the least to most probable segments of future perimeter expansion. Whereas the $fline_c$ does not necessarily lead to perimeter expansion (e.g., indicates smoldering or natural/human barriers), the $fline_r$ requires knowledge of future perimeters but offers a more precise estimate of where the perimeter expanded. The $fline_r$ is a stricter definition of the active fire line, more similar in length to $fline_c$ at high confidence thresholds. The $fline_r$ can be used for retrospective analysis to assess the drivers and barriers of fire growth."

Line 262: "Thus, fline_r is more useful for studying the trends and behaviors of historical fires." The GOFER dataset only covers fires from 2018-2020 and is therefore inherently historical, so it is not clear why fline_c is provided ("cfinelen" in Table 2).

The $fline_c$ is most similar to the method FEDS uses to produce active fire lines. It can be derived without knowing the future fire perimeter, so it can be used to test predictive modeling frameworks and evaluated with other satellite-derived products. We have revised this paragraph to clarify the use of $fline_c$ for testing predictive models and $fline_r$ for retrospective analysis:

L344-350: "In general, the $fline_c$ can be calculated in near-real-time along with perimeters and is most useful for identifying potential areas of spread along the perimeter and testing predictive models of future fire growth. The set of $fline_c$ at different confidence thresholds can be used in tandem to identify the least to most probable segments of future perimeter expansion. Whereas the $fline_c$ is not necessarily associated with perimeter expansion (e.g., indicates smoldering or natural/human barriers), the $fline_r$ requires knowledge of future perimeters but offers a more precise estimate of where the perimeter expanded. The $fline_r$ is a stricter definition of the active fire line, more similar in length to $fline_c$ at high confidence thresholds. The $fline_r$ can be used for retrospective analysis to assess the drivers and barriers of fire growth."

2.4.2 Fire spread rate

Here fire spread rate is defined in two different ways, but the authors do not provide a rationale. Please explain and, if relevant, discuss the conditions when each would be more appropriate for use. Which is at least theoretically more like to the spread rate as it would be measured on the ground?

We revise to Section 2.4.2 to the following:
L352-355: "To quantify the apparent horizontal expansion of the fire perimeter, we define the fire spread rate, in units of km/h, in two ways, as either the maximum axis of expansion ($fspread_{MAE}$) or the area-weighted expansion ($fspread_{AWE}$), between two hourly timesteps. Similar to the approach in (Benali et al., 2023), $fspread_{MAE}$ represents the partial fire spread along the longest axis, while $fspread_{AWE}$ represents the overall fire spread."

Because the fire spread rate calculation relies on the accuracy of the perimeters, a reasonable comparison should involve calculating the fire spread rate between timesteps that ground truth perimeters are available. For another method, the fire spread rate from GOFER can be calculated for each grid cell in a predefined grid. This would be more directly comparable to the spread rate measured on the ground. We add in Section 3.4.2:

L853-874: "Finally, here we rely on FEDS to evaluate the active fire lines and fire spread rates, both of which rely on the accuracy of the perimeters. More extensive evaluation and validation can be performed using aerial data and ground measurements. For example, future development of the concurrent active fire lines in GOFER could use FRP to threshold and segment active fire lines into fire intensity classes; however, this approach must account for uncertainties in the FRP calculated for saturated and low-quality fire pixels. To compare more directly to spread rates measured on the ground, the GOFER fire spread rates could be calculated for specific points or each grid cell in a predefined grid with the $fspread_{MAE}$ approach."

It may be misleading to use the term "rate of spread" for what is being calculated, though that may be what the authors believe they are calculating. For example, the (moderate) apparent growth outward of a perimeter can come about by rapid sliding downwind of air along the fire flanks, each parcel striping a bit farther out into unburned fuel. The rate any parcel of fire is spreading is then much different that the expansion rate calculated hourly normal to the fire line.

This method of calculating the fire spread rate has been used in previous studies mapping fire progression, such as Benali et al. (2023, ESSD). Please see our revision above. We have specified that we calculate the apparent horizontal outward growth of the perimeter. Satellite-derived fire spread rates cannot resolve these intricacies in physical fire behavior. We add this limitation to the discussion:

L667-668: "However, GOFER cannot be used to understand fine-scale physical fire behavior, such as spotting or convection along the fire line, due to unnatural textures resulting from spatial limitations of GOES."

2.6 Evaluation and validation

Line 298: "FEDS can take advantage of the higher spatial resolution of 375-m VIIRS detections to nearly pinpoint the exact fire locations." The more one uses VIIRS data, the less one tends to use terms indicating precision such as "pinpoint". While an excellent tool, similar problems arise when pushing VIIRS data beyond its limitations.

We revise L428-430 to: "FEDS can take advantage of the higher spatial resolution of 375-m VIIRS detections to identify fire locations more accurately than GOES, whose raw active fire detections can lead to large biases due to its much coarser spatial resolution."

Please explain how comparing at 12 hourly intervals to VIIRS data or final perimeters can be said to "validate" hourly perimeters.

We validate the final perimeters with FRAP. As FEDS is a satellite-derived product, the comparison to FEDS is not a validation of perimeters at an hourly time scale, but an evaluation of GOFER's performance across each fire's lifetime relative to a product derived from higher spatial resolution observations. To make this clearer, we have revised the text in Section 2.6. To additionally validate intermediate perimeters, we add a new validation using perimeters derived from infrared imagery provided by the U.S. Forest Service National Infrared Operations (NIROPS) Unit through the National Interagency Fire Center (NIFC) system. We retrieved and processed over 650 perimeter snapshots for the 28 fires to validate GOFER.

Section 2.2, L136-142: "NIFC provides high-resolution intermediate perimeters derived from airborne infrared (IR) imagery by trained analysts (https://data-nifc.opendata.arcgis.com/). The availability of these perimeters is sparse, varying from fire to fire and affected by cloud cover, thick smoke, and availability of flights and coverage area. For example, almost all flights are during nighttime, and some sections of the fire may not be mapped during a particular flight. We use the IR perimeters from the U.S. Forest Service National Infrared Operations (NIROPS) Unit. After filtering the NIROPS perimeters for data quality (e.g. missing metadata, small flight coverage) and matching with GOFER perimeters by the nearest hour, our reference dataset comprises over 650 snapshots across the 28 fires."

Section 2.6, L418-422: "To validate the temporal progression of GOFER perimeters, we use NIROPS perimeters derived from airborne IR imagery. We track the cumulative fire area accumulated relative to the final fire size. Because NIROPS perimeters are relatively sparse and almost all during nighttime, we additionally evaluate the performance of GOFER relative to FEDS over each fire's lifetime to check when the GOFER perimeters are relatively stable in spatial accuracy…"

Section 3.1, L475-483: "The overall temporal progression of the cumulative change in fire-wide area in GOFER agrees well with NIROPS. For example, for hours that NIROPS perimeters are available, we find a strong correlation between the change in fire area ($r = 0.86$, RMSE = 52.8 km$^2$) between NIROPS snapshots and fractions of final fire size ($r = 0.99$, RMSE = 0.05) from GOFER-Combined and NIROPS (Figure C2). High RMSE in the change in fire area mainly stems from a few instances of high bias between some snapshots in complex fires. The median absolute bias is 6.7 km$^2$, while the mean absolute bias is 16.7 km$^2$. GOFER has a median positive bias of 0.02 in the fractions of final fire size, suggesting that perimeter growth accumulates slightly earlier for GOFER than for NIROPS. As a caveat, NIROPS does not fully map the fire for some snapshots, so some areas of active growth may be missing. The discrepancies may also indicate that GOFER is unable to pick up small increments of growth later in the fire's lifetime when fire front is less active."

* Section 3.1 Evaluating the accuracy of the GOFER fire progression perimeters

Line 360: "In extreme cases, such as the Windy, Tamarack, Red Salmon Complex, and McCash fires for GOFER-Combined, we see this inability to form an initial perimeter hundreds of hours after ignition (Figure 4b)." These extreme cases comprise 4/28 = 14% of the GOFER data set. It seems appropriate to associate some sort of quality state to each GOFER fire and/or time step that would alert users to this situation.

We now include a table of the values used for early perimeter scaling for each timestep with the GOFER product.

* Section 3.2 The fire diurnal cycle derived from GOFER

Figure 9: What do the shaded area time series in the background represent?

We add to the caption: "The shaded areas represent ±1 standard deviation."

Line 407: "The lower reliability of GOES-East during the day-to-night transition period likely drives the temporal artifacts in the fire diurnal cycle in GOFER-Combined." This explanation comes off as slightly misdirected because the authors are knowingly using GOES-East near the limit of its coverage.

We have revised L586-589 to: "As GOES-East is closer to the edge of its full disk of view than GOES-West in California, GOES-East observations are inherently less reliable and more subject to issues such as sun glint and viewing zenith angles. As such, the lower reliability of GOES-East during the day-to-night transition period likely drives the temporal artifacts in the fire diurnal cycle in GOFER-Combined."

* Section 3.3 Assessing the GOFER active fire lines and fire spread rates

Lines 443-449: The agreement between FEDS and GOFER fire line (fline_c and fline_r values) is fairly poor (average correlation from 0.45 - 0.64 depending on confidence threshold). FEDS is presumably better, though I don't believe the FEDS fire lines have actually been validated, but regardless the authors should discuss the implications of this result. Are the GOFER fire line estimates good enough to improve fire spread and atmospheric transport models as the authors claim?

We have provided an additional explanation of the inconsistencies between FEDS and GOFER active fire lines:

L628-631: "Slight day-to-day differences in the retrieval times of VIIRS fire detections also affect the comparison between GOFER and FEDS active fire lines. While GOFER uses all 10-min full disk GOES images within each hour, VIIRS can only observe the state of the fire at its retrieval time, so the spatial extent and state of fire may have changed substantially at the end of the hour when GOFER and FEDS are compared."

For atmospheric transport models, the potential contribution from GOFER lies in the fire diurnal cycle information, not the active fire lines, to improve input emissions inventories. We revise Section 3.4.3 to:

L876-893: "First, GOFER can be used to improve the fire diurnal cycle for atmospheric modeling of smoke emissions. In current global fire emissions databases, the diurnal cycle is broadly generalized by land cover and generally static from day to day throughout a fire's lifetime; for example, the 3-hourly fire diurnal cycles in the Global Fire Emissions Database (GFED) are derived from historical GOES observations from 2007-2009 and implemented as climatological means based on three land cover types (van der Werf et al., 2017; Mu et al., 2011). As is evident from GOFER, however, large fires may have explosive days of growth where burning extends from the afternoon to evening. In contrast, other days with slower fire spread are generally marked only by growth during the afternoon peak. Recently, GOES observations have been merged with VIIRS observations to estimate hourly fire emissions at 3-km spatial resolution in a top-down, FRP-based approach for the Regional ABI and VIIRS fire Emissions (RAVE) product (Li et al., 2022). Similarly, for a bottom-up, burned area-based approach, the GOFER diurnal cycle of the fire-wide growth in area can be used to downscale the perimeters of select fires in existing fire progression products, such as FEDS, to hourly intervals. Second, the GOFER product can be used to build statistical and machine learning models to understand how temporal variations in weather, topography, fuels, and active fire suppression at the active fire line drive fire spread rate and fire-wide growth in area at an hourly scale. Owing to limitations in spatial resolution in both the input and output data, GOFER is most suitable for 1D time series models. For example, the GOFER product can be used to explore periods of critical stress on firefighting resources, such as in mid-August and early September of 2020 when 8-9 large fires were simultaneously active (Figure A1). Using the set of available fires in GOFER as case studies, we can identify periods when large fires are explosive or quiescent, including extreme cases when nighttime "brakes" on fire spread fail (Balch et al., 2022), causing evacuations and damaging structures."

Lines 456-463: The spread comparison is limited to the three different versions of the GOFER. The agreement is good but does not really say anything about the true accuracy of the spread rates. Also, the two different methods (MAE vs AWE) used to calculate spread rate show large differences (up to 2.5x). Is one more correct?

The accuracy of the fire spread rates depends on the accuracy of the perimeters as the calculation uses the perimeters as input. We add the following text:

Section 2.6, L405-407: "In our framework, the spatial accuracy of the perimeters directly affects that of the active fire lines and fire spread rates, both of which are derived from the perimeters. Due to limitations in high-resolution reference data, here we focus on the validation of the perimeters with FRAP and NIROPS and evaluation of active fire lines with comparisons to FEDS."

Section 3.4.2, L853-874: "Finally, here we rely on FEDS to evaluate the active fire lines and fire spread rates, both of which rely on the accuracy of the perimeters. More extensive evaluation and validation can be performed using aerial data and ground measurements. For example, future development of the concurrent active fire lines in GOFER could use FRP to threshold and segment active fire lines into fire intensity classes; however, this approach must account for uncertainties in the FRP calculated for saturated and low-quality fire pixels. To compare more directly to spread rates measured on the ground, the GOFER fire spread rates could be calculated for specific points or each grid cell in a predefined grid with the $fspread_{MAE}$ approach."

The large difference between the two spread rates is expected, since MAE represents the partial fire spread along the longest axis of expansion while AWE represents the overall fire spread between timesteps. We clarify the two methods for calculating the fire spread rate.
L352-355: "To quantify the apparent horizontal expansion of the fire perimeter, we define the fire spread rate, in units of km/h, in two ways, as either the maximum axis of expansion ($fspread_{MAE}$) or the area-weighted expansion ($fspread_{AWE}$), between two hourly timesteps. Similar to the approach in (Benali et al., 2023), $fspread_{MAE}$ represents the partial fire spread along the longest axis of expansion, while $fspread_{AWE}$ represents the overall fire spread."

* Section 3.4 Future work and applications

Line 465: "The GOFER dataset can used to address key scientific questions on fire behavior controls in California." This and some of the claims that follow seem far-fetched given the small sample size of GOFER (28 fires, mostly from 2019 and 2020) and its inherently low spatial resolution.

We revised the discussion in Section 3.4.1 to emphasize the inherent spatial limitations of GOFER but also the potential of GOFER to address these questions as the product grows to include additional fires and as the algorithm is refined with future development:

L658-673: "Here we use 28 large fires in California from 2019-2021 to test the potential of the GOFER algorithm to track the hourly progression of large wildfires using 2-km GOES active fire detections. While GOFER fills in temporal gaps in tracking fire progression, there are inherent limitations arising from the low spatial resolution of GOES observations, missed active fire detections, and potential geolocation errors in the perimeters and the active fire lines. In particular, GOFER is less reliable around water bodies and mountainous terrain. While GOES-East and GOES-West observations can be combined to increase the overall spatial accuracy of GOES-derived perimeters, we find that in California, GOFER-West is comparable to GOFER-Combined, and the use of GOES-East observations can detract from the spatial and temporal accuracy of GOFER-Combined. We expect that the suitability of the GOFER product for scientific applications, such as improving the fire diurnal cycle in emissions estimates or understanding the controls on extreme fire behavior, will grow as the algorithm is refined and additional fires are processed. However, GOFER cannot be used to understand fine-scale physical fire behavior, such as spotting or convection along the fire line, due to unnatural textures resulting from spatial limitations of GOES. Importantly, lessons learned in developing the GOFER algorithm may be applied to observations from future geostationary satellites over North and South America, such as NOAA's planned GeoXO (Geostationary Extended Observations) satellite system in the 2030s to replace the current GOES-R series with higher spatial resolution and additional bands (Adkins, 2022), and existing geostationary satellites over other regions, such as Himawari over East Asia, Equatorial Asia, and Australia and Meteosat over Europe and Africa (Hally et al., 2016; Roberts and Wooster, 2008)."

Line 475: "the GOFER dataset can be used to build temporal or spatio-temporal statistical and machine learning models to understand how variations in climate, suppression, and fuels drive fire spread rate and fire-wide growth in area." The climate reference is excessive given the short time period covered by GOFER.

L887: We have revised "climate" to "weather."

Line 478: "GOFER perimeters can be used to validate existing 3D fire spread models" At what model resolution do the authors think this might be true? The authors should keep in mind not only the low spatial resolution of GOES pixels but the non-zero error in the navigation of those pixels.

We revise the discussion in Section 3.4.1, L893-901 as follows: "For spatial analyses, GOFER could be used as a secondary product to FEDS and high-resolution perimeters from state and federal agencies. GOFER and FEDS can be used to improve the parameterization and provide a first-order validation of 3D fire spread models, such as ELMFIRE and WRF-Fire, during periods of extreme fire spread and active nighttime burning, which are often poorly estimated compared to satellite and aircraft observations (Stephens et al., 2022; Turney et al., 2023). Potential geolocation errors in GOFER and FEDS active fire lines for initializing 3D fire spread models should be accounted for, such as by perturbing the active fire lines in an ensemble approach according to the distribution of error relative to reference perimeters."

Overall:
This methodology and claims of what the work has accomplished is unsettling. It not clear what scientific use one could legitimately use the product for, what spatial and temporal "resolution" one could claim it has, or what error one can associate with any given perimeter. For example, there is so much texture to each hourly perimeter that is presented, yet (having studied several of these events in detail), nothing physical in fire behavior could support the bulges that the results show. (The unevenness of the perimeters likely arises from the error being of the order of the pixel size, as discussed elsewhere.) In contrast, the nature of the convection along the fire line has the opposite effect - to maintain a smoother fire line.

We believe that our revisions have made clear the scientific uses of the GOFER product and its methodology and limitations. In summary, GOFER is an hourly representation of fire progression using coarse geostationary satellite imagery. The spatial resolution of vector-based perimeters is typically not specified but usually referred to by the spatial resolution of the input images and spatial errors quantified relative to reference data. We have provided an additional analysis to quantify the spatial errors of GOFER perimeters relative to reference perimeters. GOFER is most suitable for improving the fire diurnal cycle in emissions inventories and understanding the temporal variability in the drivers of fire growth. As fire monitoring is an intended objective for geostationary satellites such as GOES, it is important to understand its capability, suitability, and limitations for this application. Please see our point-by-point responses above.

The revised sections on limitations and potential applications are as follows, L657-901:

"**3.4.1 Limitations**

[revised manuscript text omitted]

**Other technical corrections:**
Changes to specific fires:
For the July Complex Fire, we have expanded the spatial bounds of the search box to include the Dalton and Allen fires, which are smaller fires considered part of the complex. This change is reflected in the GOFER product and Figures 1, 4b, 8, 9, A1, and C4 and Tables 1, and C1.

For the Red Salmon Complex Fire, the GOES_UTC time was incorrectly copied in the metadata csv file in the GOFER product from the Earth Engine metadata dictionary. This change is reflected in the GOFER product and Figure C4 and Table A1.

For the Beckwourth Complex Fire, we have reprocessed the fire and manually adjusted the start time for the GOES time series to reflect the earlier start of the Dotta Fire (part of the complex). The CAL FIRE information shows a later ignition for the Beckwourth Complex Fire.

Table A1. There was a typo in the lat/lon coordinates for CZU Lightning Complex, and the previous coordinates were entered incorrectly. These coordinates were not used in the analysis and were only used in this table to illustrate some metadata for each fire. The coordinates have been corrected from Lon: -120.68 and Lat: 40.06 to Lon: -122.22 and Lat: 37.17.

Table B2: There was a typo in Table B2 on the range of the kernel size of GOFER-West. This has been corrected from 2.5-2.6 km to 2.5-2.7 km.

Figure 5. There was a typo in the top-left panel. We have corrected "exceeds" to "exceeding" and updated the figure.

We have requested a more complete set of the DINS (Damage Inspection Program) dataset of the status of structures (e.g. damaged, destroyed) within fire perimeters from CAL FIRE. The validation DINS dataset to calculate the omission error of GOFER has increased from 12 to 20 fires. There were some inconsistencies with previous data from the CAL FIRE Open Data Portal. The DINS dataset is more complete, and we redid the analysis for the 12 fires using the complete DINS data (Table C3).

In Section 2.2, we change the text accordingly to:

"For select fires (20 of 28 fires), the CAL FIRE Damage Inspection Program (DINS) database also provides the location of permanent structures inside or within 100 m of the perimeter and the level of damage sustained by each structure (accessed from the CAL FIRE Records Center at the GovQA Portal). These data are used to calculate the number of affected and destroyed structures contained by our derived fire perimeters."

**Other minor changes:**

Figure 1. We added a thin black outline to the fire detection confidence color bar to make the lighter colors clearer.

Citations have been updated to the ESSD format.

---

## Author Comment (AC2)

**Response to Referee #2**

This is a well-written paper presenting an approach to map the hourly progression of the growth of 28 large fires that burned in California during 2019–2021 using the GOES-West and GOES-east satellites. Although the GOPHER fire perimeters are generally less accurate than those detected with higher-resolution imagery, the advance here is the high hourly time resolution. Some of the implications and potential uses for the current dataset may be a bit over-sold, but the dataset will certainly increase in value presuming that it grows in the coming years, and additional value will come when lessons learned with this approach are someday applied to new satellite products with higher spatial resolution and/or other improvements on GOES.

We thank the reviewer for their helpful and constructive comments that have helped us improve our paper. Our point-by-point responses to these comments are listed below, and additional technical changes are listed at the end of our response. Our major changes include (1) restructuring the text in the methods and discussion sections for clarity, (2) adding an analysis of the spatial errors in GOFER perimeters relative to FRAP, and (3) adding a validation of intermediate GOFER perimeters with reference perimeters derived from aerial infrared imagery.

**Specific comments:**
L22: I realize that Intersection-over-Union (IoU) variable is a common metric in work related to image detection, but as a climate and fire scientist I was not aware of this metric until reading this paper and I suspect that many of the intended readers of this paper are similarly ignorant. For the Abstract, if a concise definition is not feasible, I think the main point that the fire perimeters detected in this study agree well with those from FRAP can be made without the use of the IoU variable, or the meaning of the numbers could be made more intuitive. Then, in the main text I suggest explaining the IoU and any other metrics used in this study that may not be intuitive to fire scientists who lack expertise with remote sensing and/or image detection.

Thank you for this suggestion.

In the abstract, L23-24, we add the following description of IoU: "the IoU indicates the area of overlap over the area of the union relative to the reference perimeters, in which 0 is no agreement and 1 is perfect agreement."

In methods (Section 2.3.3.3), L298-301, we revise the description of IoU to: "The IoU, or Jaccard index, is a common metric for evaluating spatial accuracy against ground truth data in object detection. Here the IoU is calculated as the area of overlap over the area of union using the fire perimeters, in which IoU = 0 indicates no agreement and IoU = 1 indicates perfect agreement."

Section 3.4 (Future work and applications): While I do see value in this work, some of the suggested applications seem unlikely. For example, with only 28 fires, low spatial resolution, and uncertainty in the specific locations of burning and fire-line position, it seems doubtful that this specific dataset will open the possibilities for new insights regarding questions about the vegetation characteristics that promote explosive or quiescent fire activity. I think some of the caveats to this section, and any impression the reader may have that GOPHER is being oversold in this section, could be addressed if the section was preceded by a section that is specifically dedicated to the inherent limitations of GOPHER.

Thank you for this feedback. We have revised Section 3.4 to first highlight the limitations of GOFER before discussing its potential applications, in response to your comments and those of reviewer 1. Section 3.4 is now divided into three subsections: 3.4.1 Limitations, 3.4.2 Future work and development, and 3.4.3 Potential applications. Sections 3.4.1 and 3.4.3, L657-673, 875-901, are revised as follows:

[revised manuscript text omitted]

We have revised the conclusion to emphasize that the GOFER dataset can be used as a case study reference in potential applications to identify periods of explosive or quiescent fire activity, rather than open opportunities for providing new insights about vegetation characteristics.

L909-914: "GOFER resolves the time dimension of fire progression mapping to hourly intervals and can identify critical, explosive periods of fire spread… Additionally, our GOFER dataset for the 28 large wildfires in California from 2019-2021 is a useful case study reference for modeling weather-human-fire relationships and improving estimates of fire emissions and smoke pollution."

**Technical corrections:**
L249: "such as"?

Thank you for pointing out the typo. We have corrected this to "such as."

**Other technical corrections:**
Changes to specific fires:
For the July Complex Fire, we have expanded the spatial bounds of the search box to include the Dalton and Allen fires, which are smaller fires considered part of the complex. This change is reflected in the GOFER product and Figures 1, 4b, 8, 9, A1, and C4 and Tables 1, and C1.

For the Red Salmon Complex Fire, the GOES_UTC time was incorrectly copied in the metadata csv file in the GOFER product from the Earth Engine metadata dictionary. This change is reflected in the GOFER product and Figure C4 and Table A1.

For the Beckwourth Complex Fire, we have reprocessed the fire and manually adjusted the start time for the GOES time series to reflect the earlier start of the Dotta Fire (part of the complex). The CAL FIRE information shows a later ignition for the Beckwourth Complex Fire.

Table A1. There was a typo in the lat/lon coordinates for CZU Lightning Complex, and the previous coordinates were entered incorrectly. These coordinates were not used in the analysis and were only used in this table to illustrate some metadata for each fire. The coordinates have been corrected from Lon: -120.68 and Lat: 40.06 to Lon: -122.22 and Lat: 37.17.

Table B2: There was a typo in Table B2 on the range of the kernel size of GOFER-West. This has been corrected from 2.5-2.6 km to 2.5-2.7 km.

Figure 5. There was a typo in the top-left panel. We have corrected "exceeds" to "exceeding" and updated the figure.

We have requested a more complete set of the DINS (Damage Inspection Program) dataset of the status of structures (e.g. damaged, destroyed) within fire perimeters from CAL FIRE. The validation DINS dataset to calculate the omission error of GOFER has increased from 12 to 20 fires. There were some inconsistencies with previous data from the CAL FIRE Open Data Portal. The DINS dataset is more complete, and we redid the analysis for the 12 fires using the complete DINS data (Table C3).
In Section 2.2, we change the text accordingly to:
"For select fires (20 of 28 fires), the CAL FIRE Damage Inspection Program (DINS) database also provides the location of permanent structures inside or within 100 m of the perimeter and the level of damage sustained by each structure (accessed from the CAL FIRE Records Center at the GovQA Portal). These data are used to calculate the number of affected and destroyed structures contained by our derived fire perimeters."

**Other minor changes:**
Figure 1. We added a thin black outline to the fire detection confidence color bar to make the lighter colors clearer.

Citations have been updated to the ESSD format.